# Towards the Difficulty for a Deep Neural Network to Learn Concepts of Different Complexities

Dongrui Liu[a*]    Huiqi Deng[a*]   Xu Cheng[a]    Qihan Ren[a]    Kangrui Wang[b]
Quanshi Zhang[a] [†]
[a]Shanghai Jiao Tong University    [b]University of Chicago

## Abstract

This paper theoretically explains the intuition that simple concepts are more likely to be learned by deep neural networks (DNNs) than complex concepts. In fact, recent studies have observed [45, 27] and proved [47] the emergence of interactive concepts in a DNN, *i.e.*, it is proven that a DNN usually only encodes a small number of interactive concepts, and can be considered to use their interaction effects to compute the inference score. Each interactive concept is encoded by the DNN to represent the collaboration between a set of input variables. Therefore, in this study, we aim to theoretically explain that interactive concepts involving more input variables (*i.e.*, more complex concepts) are more difficult to learn. Our finding clarifies the exact conceptual complexity that boosts the learning difficulty.

## 1   Introduction

Deep neural networks (DNNs) have exhibited superior performance in various tasks, but the reason for their superior performance remains an open problem. To this end, many attempts have been made to explain the representation capacity of DNNs from different perspectives. For example, Montufar et al. [35] used the number of linear response regions in a DNN to evaluate the expressive power of the DNN. Dinh et al. [13] and Petzka et al. [39] used the flatness of loss functions at minima to explain the generalization power.

In this paper, we explore a fundamental yet not well-formulated problem in terms of the representation capacity of DNNs, *i.e.*, what types of concepts are easier to be learned by DNNs. However, the core challenge of this problem is that researchers have not reached a consensus on how to define a concept encoded by DNNs. Therefore, previous findings [5, 28, 34] that *DNNs easily learn simple concepts* remain intuition or empirical observations, without a clear theoretical formulation and explanation.

Fortunately, Ren et al. [45], Li and Zhang [27] have observed and Ren et al. [47] have, for the first time, mathematically proven that the emergence of interactive concepts is a common phenomenon shared by different DNNs. ***I.e., it is proved that under some common conditions, a well-trained DNN will encode just a small number of interactive concepts for inference.*** Specifically, the interactive concept encoded by a DNN is defined as a Harsanyi interaction [18] in game theory, which represents an AND relationship between input variables in a specific set $S$. For example, as Figure 1(a) shows, the DNN encodes the co-appearance relationship between input variables (image patches) $x_1$, $x_2$, and $x_3$ to form the *mouth concept* $S = \{x_1, x_2, x_3\}$. Only when all three patches are all present, the mouth concept is triggered, and adds a numerical effect $I(S)$ to the inference score $y$. The masking of any patches (*e.g.*, $x_1$) will break this AND/co-appearance relationship of the mouth

---

[*]Equal contribution

[†]This research is done under the supervision of Dr. Quanshi Zhang. He is with the Department of Computer Science and Engineering, the John Hopcroft Center at the Shanghai Jiao Tong University, China. Correspondence to: Quanshi Zhang <zqs1022@sjtu.edu.cn>.

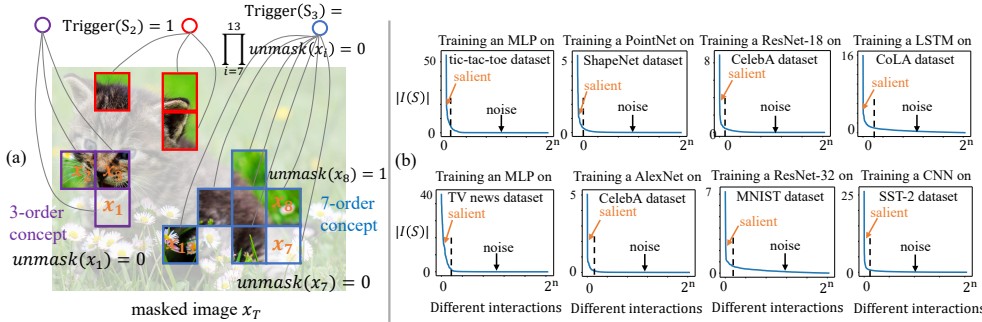

Figure 1: (a) Interactive concepts encoded by a DNN. Each interactive concept $S$ represents an AND relationship between input variables (image patches) in $S$. Masking any input variables in $S$ will deactivate the concept $S$. (b) Extensive experiments illustrate the common concept-emergence phenomenon. Various DNNs all encode very sparse interactive concepts for inference. Only a small number of interactions have salient effects $I(S)$ and are termed interactive concepts. Other interactions *w.r.t.* other $S \subseteq N$ all have almost zero effect $I(S) \approx 0$, thus being termed noisy patterns. For clarity, we sort the strength of interaction effects in a descending order.

concept, and removes the numerical effect $I(S)$, *i.e.*, making $I(S) = 0$. More crucially, it proved in [47] that **the network output $y$ on each input sample can be represented as the sum of numerical effects of such a few interactive concepts**, *i.e.*, $y = \sum_{S \in \text{ConceptSet}} I(S)$.

The mathematical proof in [47] makes the interactive concept **the first concept whose emergence in DNNs is certificated**. Furthermore, Li and Zhang [27] have discovered that the interactive concepts **have considerable transferability over different samples and strong discrimination power for classification.** In this way, we can roughly consider such interactive concepts a rigorous definition of concepts encoded by DNNs.

**Explaining the difficulty of learning complex interactive concepts.** Therefore, in this study, given the above interactive concepts, we aim to further explore the learning difficulty of these concepts, and explain that a DNN is more likely to encode simple interactive concepts. Here, we use the number of variables in an interactive concept $S$ to represent the complexity of the concept, which is also termed *the order* of the concept. Thus, low-order interactive concepts usually represent simple AND interactions among a few input variables.

Specifically, we find that low-order interactive concepts encoded by DNNs are generally more stable to inevitable noises in data. In this way, low-order interactive concepts usually exhibit consistently positive (or negative) effects on the inference score of different samples in the same category. In comparison, a high-order interactive concept is more likely to have significantly diverse effects on inference scores of different samples. This explains the reason why DNNs easily learn simple interactive concepts.

In addition, we discover that our research may provide a new perspective to explain previous findings/understandings of the conceptual complexity [5, 34, 63, 28]. Besides, the conceptual complexity can directly explain the adversarial robustness of a DNN. Thus, our study is of considerable value in explaining the representation capacity of a DNN.

**Interactive concepts vs. cognitive concepts.** Although the Harsanyi interactive concepts visualized by [27] seem partially aligned with cognitive concepts to some extent, we do not think such interactive concepts fit humans' cognition [24, 53]. For example, the recognition of a *big ball concept* and that of a *small ball concept* have almost the same cognitive complexity for humans. In comparison, it is more difficult for the DNN to recognize a large ball, because it has to use a high-order interaction that contains a lot of pixels to examine whether all pixels within the ball have the same color. Thus, this study clarifies the exact formulation for the conceptual complexity on interactive concepts that boosts the learning difficulty, which is different from the cognitive difficulty.

## 2 Explainable AI (XAI) theories based on game-theoretic interactions

Explaining DNNs based on a system of game-theoretic interactions becomes an emerging direction in recent years. Specifically, such a system aims to tackle the following challenges in XAI.

• *Defining, extracting, and quantifying interactive concepts encoded by DNNs.* Quantifying the interactions between different input variables [57, 58] is a new perspective to formulate the knowledge encoded by a DNN. Based on game theory, Zhang et al. [66, 68, 67] introduced multi-variate interaction and multi-order interaction to quantify the interactions encoded by the DNN. Ren et al. [45] empirically observed and Ren et al. [47] further proved that a DNN usually just encoded a small number of salient interactions. Ren et al. [44] defined the optimal baseline value for computing the Shapley value based on these interactions. Li and Zhang [27] found that salient interactions usually have high transferability over different samples and DNNs. These findings showed that these salient interactions could be viewed as interactive concepts encoded by a DNN.

• *Using interactive concepts to explain/measure the representation power of DNNs.* The representation power of DNNs has been explained by game-theoretic multi-order interaction, including generalization power [67, 70], adversarial robustness [45, 70], and adversarial transferability [60]. Besides, Cheng et al. [8] used interactions to investigate how shapes and textures were encoded in DNNs. Cheng et al. [9] found that salient interactions often represented prototypical patterns encoded by DNNs. Deng et al. [11] derived that the DNN was less likely to encode mid-order interactions. In comparison, Ren et al. [46] proved that the Bayesian neural network was less likely to encode high-order interactions, thereby obtaining good adversarial robustness.

• *Unifying the common underlying mechanism shared by previous empirical findings.* Interactions have superior representation power, and we find that many previous studies can be re-explained from the perspective of interactions. Deng et al. [12] proved that fourteen attribution methods could be reformulated and considered as a certain allocation of interaction effects. Besides, Zhang et al. [69] proved that reducing the interactions was the common underlying mechanism shared by twelve previous studies for improving adversarial transferability.

## 3 Explaining simple interactive concepts are easy to learn

### 3.1 Preliminaries: sparse interactive concepts emerge in DNNs

It is generally believed that the learning of DNNs is a fitting problem between the ground-truth label and the model output, instead of explicitly learning specific concepts like graphical models. However, Ren et al. [45], Li and Zhang [27] have empirically observed and Ren et al. [47] have mathematically proven the counter-intuitive emergence of concepts in DNNs, *i.e.*, **it is proven that under a set of common conditions, a DNN encodes just *a small number of* interactive concepts for inference.**

Specifically, given a pre-trained DNN $v$ and an input sample $\boldsymbol{x} = [x_1, \ldots, x_n]$ with $n$ variables indexed by $N = \{1, \ldots, n\}$, $v(\boldsymbol{x}) \in \mathbb{R}$ denotes a scalar output of the DNN[3]. Each interaction in [45, 27, 47] is defined as Harsanyi dividend (or Harsanyi interaction) [18] in game theory, which represents a collaboration (AND relationship) between input variables in a specific set $S$ ($S \subseteq N$). For example, as Figure 1(a) shows, the co-appearance of image patches forms a mouth interaction $S = \{x_1, x_2, x_3\}$. Only when these three patches are all present, the mouth interaction will be triggered, and make a certain interaction effect $I(S)$ on the network output. The absence (masking) of any patches of $x_1$, $x_2$, and $x_3$ will deactivate the mouth interaction and remove the interaction effect, *i.e., $I(S) = 0$*. The interaction effect $I(S|\boldsymbol{x})$ on the input sample $\boldsymbol{x}$ is computed as follows.

$$I(S|\boldsymbol{x}) = \sum\nolimits_{T \subseteq S} (-1)^{|S|-|T|} \cdot v(\boldsymbol{x}_T), \tag{1}$$

where $\boldsymbol{x}_T$ denotes the masked input sample, when we mask input variables in $N \setminus T$ and keep variables in $T$ unchanged. Please see more properties of the Harsanyi dividend in the supplementary material.

Theoretically, there are $2^n$ potential subsets $S$ of input variables ($S \in 2^N = \{S|S \subseteq N\}$). The proven concept-emergence phenomenon refers to that **only a small number of subsets $S \in \Omega_{\text{salient}} \subseteq 2^N$ make salient interaction effects $I(S|\boldsymbol{x})$ on the network output, and can be considered as *interactive concepts.*** Interaction effects of all other subsets are close to zero ($I(S|\boldsymbol{x}) \approx 0$), which can be considered as ignorable *noisy patterns*. Therefore, **the network output $v(\boldsymbol{x})$ can be well approximated by interaction effects of a small number of interactive concepts,** *i.e.*,

$$v(\boldsymbol{x}) = \sum\nolimits_{S \subseteq N} I(S|\boldsymbol{x}) \approx \sum\nolimits_{S \in \Omega_{\text{salient}}} I(S|\boldsymbol{x}) \tag{2}$$

---

[3]Here, $v(\boldsymbol{x}) \in \mathbb{R}$ can be implemented as either a scalar output of the DNN or a dimension of an output vector (*e.g.*, the confidence score of classifying the sample $\boldsymbol{x}$ to the ground-truth category $v(\boldsymbol{x}) = \log \frac{p(y=y_{\text{truth}}|\boldsymbol{x})}{1-p(y=y_{\text{truth}}|\boldsymbol{x})}$).

We also conduct experiments to illustrate the concept-emergence phenomenon on various DNNs, including multi-layer perceptrons (MLPs), long short-term memory (LSTM), AlexNet [23], ResNet [19], convolutional neural networks (CNNs), and PointNet [41] trained on different types of data, including tabular data (UCI dataset [6]), natural language data (CoLA [61] and SST-2 [56]), image data (MNIST [26] and CelebA [31]), and point cloud data (ShapeNet [65]). We follow experimental settings in [45, 27], and Figure 1(b) shows that interactive concepts encoded by a DNN are usually indeed sparse.

**Theorem 1.** *Given an input sample $x \in \mathbb{R}^n$ with $n$ variables, there are $2^n$ different masked samples $x_T$ w.r.t. all potential subsets $T \subseteq N$. Then, [45] has proven that*

$$\forall T \subseteq N, \ v(x_T) = \sum_{S \subseteq T} I(S|x) \approx \sum_{S \in \Omega_{\text{salient}} \& S \subseteq T} I(S|x). \tag{3}$$

Theorem 1 means that we can use a small number of interactive concepts in $\Omega_{\text{salient}}$ to well approximate network outputs on anyone $x_T$ of the $2^n$ masked samples.

**Trustworthiness of interactive concepts.** The mathematical proof in [47] makes $I(S)$ become the first concepts whose emergence in DNNs is certificated. Equations (2) and (3) further guarantee that interactive concepts can faithfully explain the output of DNNs. In addition, Li and Zhang [27] have discovered that (i) interactive concepts are transferable across different samples; (ii) interactive concepts are discriminative, *i.e.*, if a concept $S$ has salient interaction effects $I(S)$ on a set of samples, then the concept tends to push these samples to be classified towards the same category. In this way, we can roughly consider such interactive concepts a relatively rigorous definition of concepts.

**Complexity (order) of interactive concepts.** The complexity of an interactive concept $S$ is defined as the number of input variables involved in $S$, which is also termed *the order* of the interactive concept, *i.e.*, complexity$(S) = $ order$(S) = |S|$. Thus, low-order interactive concepts usually represent simple AND relationships among a few input variables. In comparison, high-order interactive concepts often refer to as relatively complex AND relationships among a large number of input variables.

## 3.2 Simple interactive concepts in data are more stable and easier to learn

Unlike other interaction metrics [16, 32, 57], sparse interactive concepts are certificated to emerge in DNN, and can faithfully represent the inference score of DNNs. Therefore, in this study, we aim to further explore the learning difficulty of these certificated concepts. Specifically, we theoretically explain that a DNN is more likely to encode simple (*i.e.*, low-order) interactive concepts.

In this subsection, we show that low-order interactive concepts encoded by DNNs are generally more stable to inevitable noises in data, compared to high-order interactive concepts. Specifically, we derive an approximate analytical solution to the variance (instability) of interactive concepts' effects *w.r.t.* data noise, and show that the variance (instability) increases along the order of concepts in an exponential manner.

**Theorem 2** (proven in the supplementary material). *Given a neural network $v$ and an arbitrary input sample $x' \in \mathbb{R}^n$, the network output can be decomposed by using the Taylor expansion i.e., $v(x') = \sum_{S \subseteq N} \sum_{\pi \in Q_S} U_{S,\pi} \cdot J(S, \pi|x')$. In this way, according to Equation (1), the Harsanyi interaction effect $I(S|x')$ on the sample $x'$ can be reformulated as follows.*

$$I(S|x') = \sum_{\pi \in Q_S} U_{S,\pi} \cdot J(S, \pi|x'). \tag{4}$$

*Here, $J(S, \pi|x') = \prod_{i \in S} \left( \text{sign}(x'_i - b_i) \cdot \frac{x'_i - b_i}{\tau} \right)^{\pi_i}$ denotes a Taylor expansion term of the degree $\pi$, where the degree $\pi \in Q_S = \{[\pi_1, \ldots, \pi_n] | \forall i \in S, \pi_i \in \mathbb{N}^+; \forall i \notin S, \pi_i = 0\}$ and $b_i$ is the baseline value to mask the input variable $x_i$. In addition, $U_{S,\pi} = \frac{\tau^m}{\prod_{i=1}^n \pi_i!} \frac{\partial^m v(x'_\emptyset)}{\partial x_1^{\pi_1} \ldots \partial x_n^{\pi_n}} \cdot \prod_{i \in S} [\text{sign}(x'_i - b_i)]^{\pi_i}$, where $x'_\emptyset$ denotes the sample whose input variables are all masked. $m = \sum_{i=1}^n \pi_i$.*

Theorem 2 rewrites the Harsanyi interaction effect $I(S|x)$ of each concept from a new perspective, to facilitate the analysis of the instability of interactive concepts. The derivation of Theorem 2 is inspired by [12, 70, 46], in which the Harsanyi interaction effect is re-written as the sum of the Taylor interaction effects. Therefore, in this study, we define a new baseline value $b_i$ to further extend the finding in [12] and obtain Theorem 2, which significantly simplifies the further proof. Specifically, the new baseline value $b_i$ is set as follows to represent the masked state of the $i$-th input variable. *i.e.*,

$x_i \leftarrow b_i$. Previous studies usually set $b_i = \mu_i = \mathbb{E}_{\boldsymbol{x}}[x_i]$ as the average value over different samples to represent the masked state [4, 12]. However, we consider that pushing the input variable $x_i$ to move a large distance $\tau \in \mathbb{R}$ towards $\mu_i$ has been significant enough to remove its information. To this end, we set $b_i = x_i - \tau^4$, if $x_i > \mu_i$; else, $b_i = x_i + \tau^4$. The above new setting ensures comparable perturbation magnitudes over different dimensions.

Next, we analyze the variance (instability) of the interaction effect $I(S|\boldsymbol{x}' = \boldsymbol{x} + \boldsymbol{\epsilon})$ when we add a Gaussian perturbation $\boldsymbol{\epsilon} \sim \mathcal{N}(\mathbf{0}, \sigma^2 \boldsymbol{I})$ to the input sample $\boldsymbol{x}$. Here, the Gaussian perturbation $\boldsymbol{\epsilon} \in \mathbb{R}^n$ is considered as a rough representation of inevitable variations in data, *e.g.*, shape deformation and object rotation variations in image classification. However, such variations are quite difficult to formulate, so we use a Gaussian perturbation as a rough representation. Fortunately, we find that our reformulation of interactions in Theorem 2 enables us to directly apply the finding in [46] to prove the variance (instability) of the interaction effect $I(S|\boldsymbol{x}' = \boldsymbol{x} + \boldsymbol{\epsilon})$.

**Theorem 3.** *Let us add a Gaussian perturbation $\boldsymbol{\epsilon} \sim \mathcal{N}(\mathbf{0}, \sigma^2 \boldsymbol{I})$ to the input sample $\boldsymbol{x}$. Let us first consider the case with the lowest degree $\hat{\boldsymbol{\pi}} = [\hat{\pi}_1, \ldots, \hat{\pi}_n] \in Q_S$, satisfying that $\forall i \in S, \hat{\pi}_i = 1; \forall i \notin S, \hat{\pi}_i = 0$. The mean and variance of $J(S, \hat{\boldsymbol{\pi}}|\boldsymbol{x} + \boldsymbol{\epsilon})$ over the Gaussian perturbation $\boldsymbol{\epsilon}$ are given as*

$$\mathbb{E}_{\boldsymbol{\epsilon}}[J(S, \hat{\boldsymbol{\pi}}|\boldsymbol{x} + \boldsymbol{\epsilon})] = 1, \quad \mathrm{Var}_{\boldsymbol{\epsilon}}[J(S, \hat{\boldsymbol{\pi}}|\boldsymbol{x} + \boldsymbol{\epsilon})] = \left(1 + \left(\frac{\sigma}{\tau}\right)^2\right)^{|S|} - 1. \tag{5}$$

*Furthermore, for the more general case with an arbitrary degree $\boldsymbol{\pi} \in Q_S = \{[\pi_1, \cdots, \pi_n] | \forall i \in S, \pi_i \in \mathbb{N}^+; \forall i \notin S, \pi_i = 0\}$, the mean and variance of $J(S, \boldsymbol{\pi}|\boldsymbol{x} + \boldsymbol{\epsilon})$ are computed as*

$$\mathbb{E}_{\boldsymbol{\epsilon}}[J(S, \boldsymbol{\pi}|\boldsymbol{x} + \boldsymbol{\epsilon})] = \mathbb{E}_{\boldsymbol{\epsilon}}[\prod_{i \in S}(1 + \frac{\epsilon_i}{\tau})^{\pi_i}], \quad \mathrm{Var}_{\boldsymbol{\epsilon}}[J(S, \boldsymbol{\pi}|\boldsymbol{x} + \boldsymbol{\epsilon})] = \mathrm{Var}_{\boldsymbol{\epsilon}}[\prod_{i \in S}(1 + \frac{\epsilon_i}{\tau})^{\pi_i}]. \tag{6}$$

Theorem 3 shows the (variance) instability of interactive concepts of different complexities (orders) *w.r.t.* data variations. It indicates that the variance of $J(S, \boldsymbol{\pi}|\boldsymbol{x} + \boldsymbol{\epsilon})$ roughly increases along with the order$(S) = |S|$ in an exponential manner. Similar conclusions are also introduced in [70]. Furthermore, according to Equation (4), the interaction effect $I(S|\boldsymbol{x} + \boldsymbol{\epsilon})$ of the concept $S$ can be represented as the weighted sum of $J(S, \boldsymbol{\pi}|\boldsymbol{x} + \boldsymbol{\epsilon})$, and coefficients $U_{S,\boldsymbol{\pi}}$ *w.r.t.* different orders $s$ and degrees $\boldsymbol{\pi}$ are usually chaotic. Hence, *we can roughly consider that the variance (instability) of the interaction effect $I(S|\boldsymbol{x} + \boldsymbol{\epsilon})$ increases along with the order $|S|$ exponentially, as well.* In other words, **compared to high-order concepts, low-order concepts are much more stable to slight input perturbations.**

**Experimental verification.** Here, we conduct experiments to verify whether the variance (instability) of interaction effects indeed increased along with the order in an approximately exponential manner. Specifically, to mimic variations in the data, we add Gaussian perturbations $\boldsymbol{\epsilon} \sim \mathcal{N}(\mathbf{0}, 0.02^2 \boldsymbol{I})$ to each training sample. Then, we compute the average mean $E^{(s)} = \mathbb{E}_{\boldsymbol{x} \in \boldsymbol{X}}[\mathbb{E}_{S \subseteq N, |S| = s}[|\mathbb{E}_{\boldsymbol{\epsilon}}[I(S|\boldsymbol{x} + \boldsymbol{\epsilon})]|]]$ and the average variance $V^{(s)} = \mathbb{E}_{\boldsymbol{x} \in \boldsymbol{X}}[\mathbb{E}_{S \subseteq N, |S| = s}[\mathrm{Var}_{\boldsymbol{\epsilon}}[I(S|\boldsymbol{x} + \boldsymbol{\epsilon})]]]$ of interaction effects of the $s$-th order concepts *w.r.t.* Gaussian perturbations. For verification, we calculate such metrics on DNNs trained for image classification and DNNs trained on tabular data. We train AlexNet [23], VGG-11 [55], ResNet-18/20 [19] on the CIFAR-10 dataset [22] and the Tiny-ImageNet dataset [25], respectively. We also train a five-layer MLP [45] on the UCI census dataset (namely *census dataset*) and the UCI TV news dataset (namely *TV news dataset*) [6], respectively. In addition, considering the computational cost of $I(S|\boldsymbol{x} + \boldsymbol{\epsilon})$ in Equation (1) is intolerable if each pixel is considered as an input variable, we divide images into $8 \times 8$ patches [45] to calculate $I(S|\boldsymbol{x} + \boldsymbol{\epsilon})$. Please see the supplementary material for more implementation details.

Figure 2 shows that the variance (instability) $V^{(s)}$ of interaction effects increases in an exponential manner along with the order $s$ of the interactive concept. In addition, the relative stability $E^{(s)}/\sqrt{V^{(s)}}$ of interaction effects decreases significantly along with the order $s$. Such phenomenon successfully verifies findings in Theorem 3.

### 3.2.1 Formulating the conceptual learning as a linear regression problem

Note that, Li and Zhang [27] have discovered that interactive concepts have high transferability over samples in the same category, *i.e.*, most interactive concepts extracted from different samples may

---

[4]We need to avoid the case of over-perturbation, by setting $b_i \leftarrow \max(b_i, \mu_i)$ if $x_i > \mu_i$; $b_i \leftarrow \min(b_i, \mu_i)$, otherwise. Fortunately, such cases are uncommon in real applications, so we ignore such settings in the following theoretical analysis.

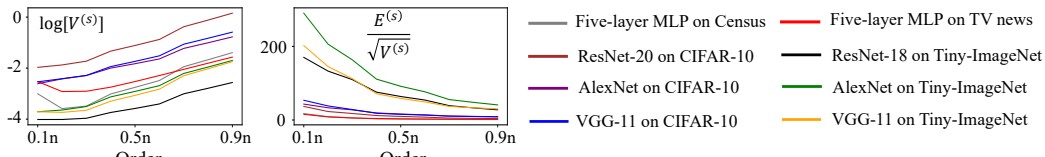

Figure 2: The logarithm of the variance of interaction effects $\log[V^{(s)}]$ and the stability of interaction effects (measured by $E^{(s)}/\sqrt{V^{(s)}}$). The variance $V^{(s)}$ of interaction effects increases along with the order of interactive concepts exponentially. The stability of effects decreases along with the order.

all belong to the same subset $\Omega_c \subseteq \Omega_{\text{salient}}$. Therefore, we can consider the interactive concepts as common knowledge learned by the DNN, and we aim to analyze the difficulty of learning each interactive concept with different complexity.

To facilitate the analysis, we first simplify the conceptual learning as a linear regression problem. Specifically, we first rewrite the interaction effect of an interactive concept $S$. Given an input sample $\boldsymbol{x}$, according to Equation (4), the interaction effect of the concept $S$ on the sample $\boldsymbol{x}'$ (obtained by applying some transformations on $\boldsymbol{x}$), $I(S|\boldsymbol{x}')$, can be rewritten as

$$I(S|\boldsymbol{x}') = U_S \cdot C_S(\boldsymbol{x}'), \tag{7}$$

where the constant $U_S = I(S|\boldsymbol{x})$ denotes the interaction effect on the sample $\boldsymbol{x}$, and the function for the triggering state on the transformed sample $\boldsymbol{x}'$ is given as $C_S(\boldsymbol{x}') = \sum_{\boldsymbol{\pi} \in Q_S} U_{S,\boldsymbol{\pi}} J(S, \boldsymbol{\pi}|\boldsymbol{x}')/U_S$.

**Theorem 4.** *Given an arbitrarily masked sample $\boldsymbol{x}_T (T \subseteq N)$, the function $C_S(\boldsymbol{x}_T)$ defined above can be computed as the binary triggering state of the concept $S$ in the sample $\boldsymbol{x}_T$.*

$$\forall\, T \subseteq N,\ C_S(\boldsymbol{x}_T) = \prod_{i \in S} A_i(\boldsymbol{x}_T) = \mathbb{1}(S \subseteq T), \tag{8}$$

*where $A_i(\boldsymbol{x}_T) \in \{0, 1\}$ denotes whether the variable $x_i$ is present $A_i(\boldsymbol{x}_T) = 1$ or being masked $A_i(\boldsymbol{x}_T) = 0$ in the sample $\boldsymbol{x}_T$.*

The function $C_S(\boldsymbol{x}_T)$ represents the triggering state of the concept $S$ under an arbitrarily masking condition $\forall\, T \subseteq N$. Only when all variables in $S$ are present under the masking condition $T$, the concept $S$ is triggered $C_S(\boldsymbol{x}_T) = 1$. If any of variables in $S$ is masked, then the concept $S$ will not be triggered $C_S(\boldsymbol{x}_T) = 0$, yielding zero interaction effect $I(S|\boldsymbol{x}_T) = 0$. Theorem 4 shows that given a masked sample $\boldsymbol{x}_T$, the concept $S$ is only triggered when the sample $\boldsymbol{x}_T$ contains all variables in $S$, *i.e.*, $T \supseteq S$.

Thus, inspired by [46], we can extend the conclusion to a continuous version that explains the output of the DNN as a **linear regression of very few salient interactive concepts** based on Equation (3).

$$v(\boldsymbol{x}') = \sum_{S \subseteq N} U_S \cdot C_S(\boldsymbol{x}') \approx \sum_{S \in \Omega_{\text{salient}}} U_S \cdot C_S(\boldsymbol{x}'), \tag{9}$$

where the triggering state $C_S(\boldsymbol{x}')$ of each interactive concept $S$ can be considered as an input dimension of the linear function, which reflects whether the input sample $\boldsymbol{x}'$ contains the concept $S$.

Therefore, the absolute value of the weight coefficient $U_S$ in Equation (9) can be viewed as *the strength of the DNN encoding the interactive concept $S$*. Considering the sparsity of interactive concepts discussed in Section 3.1, most interactive concepts have ignorable coefficients $U_S \approx 0$, and not so many concepts $S$ have large absolute value of $|U_S|$. Thus, we can consider that the DNN only learns a small number of salient interactive concepts.

### 3.2.2 Explaining the learning difficulty of concepts

Equation (9) in the previous subsection enables us to understand a DNN for the classification task as a pseudo-linear function. Then, if an input feature dimension has a stable value (*i.e.*, the triggering state $C_S$ of an interactive concept $S$ stably being present/absent) across all samples in the same category, then we consider this feature dimension (*i.e.*, the concept) is easy to learn. In comparison, when we extract the same concept from different samples in the same category, if this concept exhibits inconsistent interaction effects (*e.g.*, such a concept does not consistently present or absent over different samples), then this feature dimension (*i.e.*, the concept) is hard to learn.

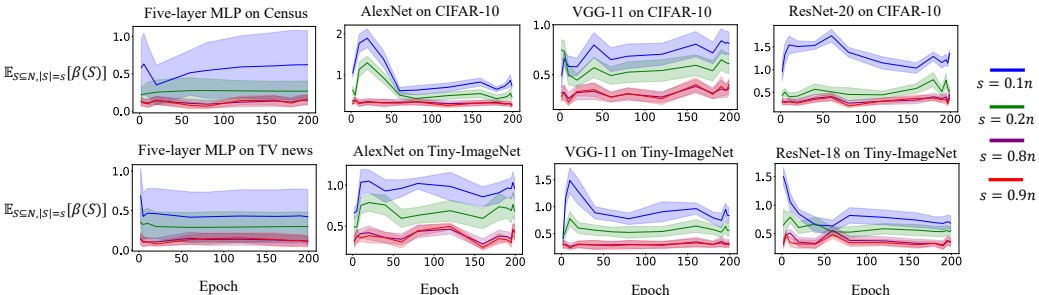

Figure 3: Consistency $\beta(S)$ of $s$-order interactive concepts. The curve shows the mean consistency $\mathbb{E}_{S\subseteq N,|S|=s}[\beta(S)]$ over different interactive concepts of the $s$-th order, and the shade indicates the standard deviation $\text{Std}_{S\subseteq N,|S|=s}[\beta(S)]$. Effects of low-order concepts are more consistent than those of high-order concepts.

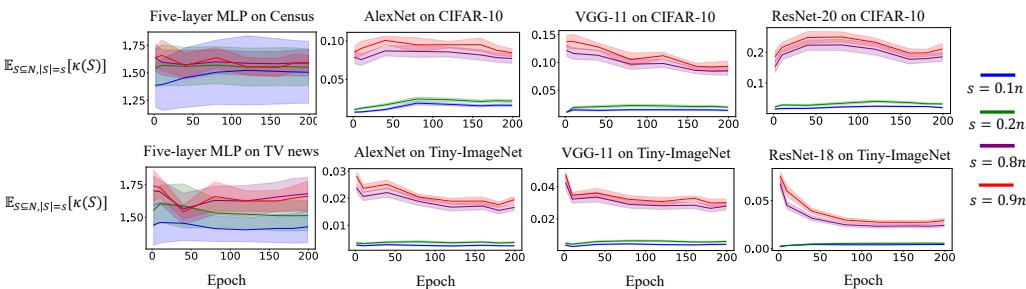

Figure 4: Instability $\kappa(S)$ of $s$-order interactive concepts to data variations. The curve shows the average instability $\mathbb{E}_{S\subseteq N,|S|=s}[\kappa(S)]$ over different interactive concepts of the $s$-th order, and the shade represents the standard deviation $\text{Std}_{S\subseteq N,|S|=s}[\kappa(S)]$. Effects of high-order concepts are more sensitive to data variations than those of low-order concepts.

To analyze the consistency/stability of the concept *w.r.t* data variations, we use small perturbations $\epsilon$ to represent various inevitable variations in the data, such as shape deformation and object rotation variations. Theorem 3 shows that high-order interactive concepts are much more unstable to inevitable variations in the data than low-order interactive concepts. This makes high-order concepts more likely to be influenced by data variations and less likely to be consistently present or absent in samples of the same category, which boosts the learning difficulty.

**Experimental verification.** We conduct experiments to verify the claim that high-order interactive concepts are less stably extracted under data variations than low-order interaction. We use the following two metrics to evaluate the each interactive concept $S$ (*i.e.,* each single input dimension of the above linear function) of a certain order. Specifically, we use the first metric $\beta(S) = \mathbb{E}_c\big[|\mathbb{E}_{\boldsymbol{x}\in\boldsymbol{X}_c}[I(S|\boldsymbol{x})]|/\text{Std}_{\boldsymbol{x}\in\boldsymbol{X}_c}[I(S|\boldsymbol{x})]\big]$ to measure the relative consistency of the interactive concept appearing over different input samples in a certain category $c$. Here, $\boldsymbol{X}_c$ denotes a set of training samples belonging to the category $c$, and $\text{Std}_{\boldsymbol{x}\in\boldsymbol{X}_c}[I(S|\boldsymbol{x})]$ indicates the standard deviation of $I(S|\boldsymbol{x})$ over different input samples. A large $\beta(S)$ value means that the interactive concept $S$ in all samples of the category $c$ has similar/consistent effects $I(S|\boldsymbol{x})$. It is easier for a DNN to learn such a consistent concept $S$.

In addition, we use another metric $\kappa(S)$ to verify whether the interaction effect of the high-order interactive concept is usually less stably extracted than those of the low-order interactive concept. To this end, the metric $\kappa(S) = \mathbb{E}_{\boldsymbol{x}\in\boldsymbol{X}}[\mathbb{E}_\epsilon[|I(S|\boldsymbol{x}+\epsilon) - I(S|\boldsymbol{x})|]]/\mathbb{E}_{\boldsymbol{x}\in\boldsymbol{X}}\big[|I(S|\boldsymbol{x})|\big] = \mathbb{E}_{\boldsymbol{x}\in\boldsymbol{X}}[\mathbb{E}_\epsilon[|C_S(\boldsymbol{x}+\epsilon) - C_S(\boldsymbol{x})|]]/\mathbb{E}_{\boldsymbol{x}\in\boldsymbol{X}}\big[|C_S(\boldsymbol{x})|\big]$ measures the relative instability of the interactive concept $S$ to the inevitable slight variations $\epsilon$ in the data. Here, $C_S(\boldsymbol{x}+\epsilon) = I(S|\boldsymbol{x}+\epsilon)/U_S$ computed in Equation (7) denotes the trigger state of the interactive concept $S$ on the sample $\boldsymbol{x}' = \boldsymbol{x} + \epsilon$.

To this end, we use DNNs and experimental settings in the *experimental verification* paragraph of Section 3.2 for evaluation. Figure 3 and Figure 4 show the change of the average consistency $\beta(S)$ and the average instability $\kappa(S)$ of $s$-order interactive concepts through the learning process, respectively. At each training epoch, low-order concepts usually obtain higher consistency $\beta(S)$ and

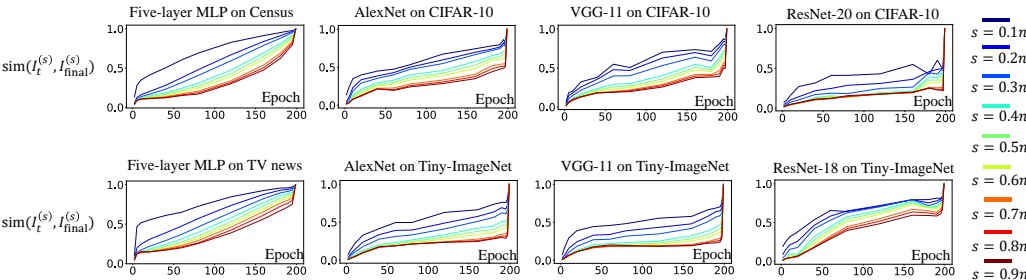

Figure 5: The weighted Jaccard similarity $\text{sim}(\boldsymbol{I}_t^{(s)}, \boldsymbol{I}_{\text{final}}^{(s)})$ between $s$-order interactive concepts learned after the intermediate epoch $\boldsymbol{I}_t^{(s)}$ and those learned after the final epoch $\boldsymbol{I}_{\text{final}}^{(s)}$. Low-order concepts usually have higher Jaccard similarity during the learning process, which indicates that DNNs first learn low-order concepts and then gradually learn more about high-order concepts.

lower instability $\kappa(S)$ than high-order concepts. It means that low-order concepts usually are more consistent and more stable. Thus, this experiment explains the reason why low-order interactive concepts are easier to learn.

### 3.2.3 Fast learning of low-order concepts

In this subsection, we illustrate the phenomenon that low-order interactive concepts are usually learned faster than high-order concepts, as a support for the claim that low-order interactive concepts are easier to be learned. We have theoretically proven in Section 3.2 and experimentally verified that low-order interactive concepts are more consistently present or consistently absent in different samples of the same category, which makes low-order interactive concepts easier to be learned. Then, this experiment is conducted to check whether low-order concepts are really learned faster than high-order concepts.

Specifically, we examine whether interactive concepts encoded by the finally-learned DNN $v_{\text{final}}(\boldsymbol{x})$ have already been encoded by the DNN that is $v_t(\boldsymbol{x})$ trained after $t$ epochs. If so, we consider such interactive concepts are learned fast. Specifically, let $\boldsymbol{I}_t^{(s)} = [I_t(S_1|\boldsymbol{x}), \cdots, I_t(S_d|\boldsymbol{x})]^\top \in \mathbb{R}^d$ denote a vector of interaction effects for all $d = \binom{n}{s}$ interactive concepts of $s$-order. Then, we compute the Jaccard similarity between $s$-order interactive concepts encoded by the DNN $v_{\text{final}}(x)$ and those encoded by the DNN $v_t(x)$, *i.e.*, $\text{sim}(\boldsymbol{I}_t^{(s)}, \boldsymbol{I}_{\text{final}}^{(s)}) = \| \min(\widetilde{\boldsymbol{I}}_t^{(s)}, \widetilde{\boldsymbol{I}}_{\text{final}}^{(s)}) \|_1 / \| \max(\widetilde{\boldsymbol{I}}_t^{(s)}, \widetilde{\boldsymbol{I}}_{\text{final}}^{(s)}) \|_1$ and $\| \cdot \|_1$ denotes L1-norm. We extend the $d$-dimensional vector $\boldsymbol{I}_t^{(s)}$ into a $2d$-dimensional vector with non-negative elements $\widetilde{\boldsymbol{I}}_t^{(s)} = [(\boldsymbol{I}_t^{(s),+})^\top, (\boldsymbol{I}_t^{(s),-})^\top]^\top = [\max(\boldsymbol{I}_t^{(s)}, 0)^\top, -\min(\boldsymbol{I}_t^{(s)}, 0)^\top]^\top \in \mathbb{R}^{2d}$. Similarly, we compute $\widetilde{\boldsymbol{I}}_{\text{final}}^{(s)}$ with non-negative elements based on $\boldsymbol{I}_{\text{final}}^{(s)}$. In this way, a large similarity $\text{sim}(\boldsymbol{I}_t^{(s)}, \boldsymbol{I}_{\text{final}}^{(s)})$ at an earlier epoch $t$ indicates that interactive concepts are easier to be learned.

To this end, let us use DNNs and experimental settings introduced in the *experimental verification* paragraph of Section 3.2 for evaluation. Figure 5 shows that DNNs first learn low-order interactive concepts, and then learn high-order interactive concepts. Such a phenomenon verifies the conclusion that a DNN easily learns low-order (simple) interactive concepts.

## 4 Explaining findings in previous studies

### 4.1 Explaining adversarial robustness

Previous study [43] has discovered that low-order interactions are more robust to adversarial attacks than high-order interactions. Notice that their high-order (low-order) interactions are different from, but highly related to our high-order (low-order) interactive concepts. Specifically, our interactive concepts can be explained as elementary components for such multi-order interactions in [43]. Please see the supplementary material for the proof. *Therefore, from this perspective, our conclusion that "low-order interactive concepts are easy to learn" can also explain how a DNN encodes concepts of different adversarial robustness.*

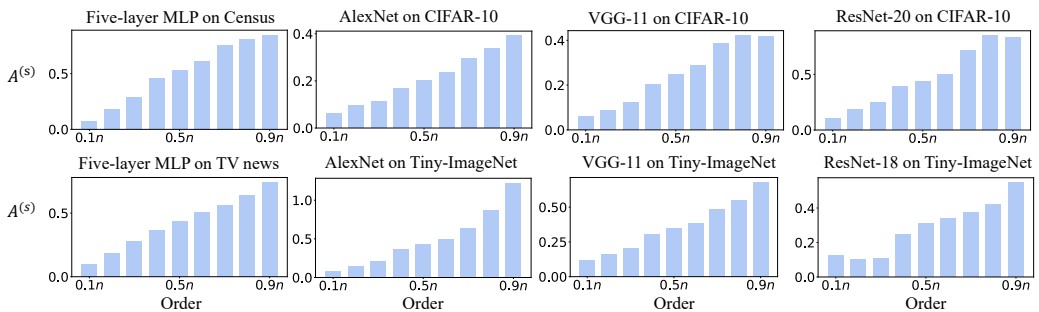

Figure 6: Average adversarial sensitivity $A^{(s)}$ of $s$-order interactive concepts to adversarial perturbations. Low-order interactive concepts are usually much less sensitive to adversarial attacks than high-order interactive concepts.

**Can we use interactive concepts in this paper to verify the heuristic findings of adversarial robustness in [43]?** According to Equation (2), we can represent the inference logic of a DNN by a set of interactive concepts, *i.e.*, $v(\boldsymbol{x}) \approx \sum_{S \in \Omega_{\text{salient}}} I(S|\boldsymbol{x})$. In this way, we conduct experiments to evaluate the sensitivity of each interactive concept $S$ to adversarial perturbations. The symbolic conceptual representation of interactive concepts allows us to measure the adversarial robustness in a more direct way. That is, if the effect of an interactive concept does not significantly change under adversarial attacks, we consider this interactive concept is robust to adversarial attacks. In contrast, if the effect of an interactive concept changes significantly under adversarial attacks, we consider this interactive concept is sensitive to adversarial attacks.

To this end, we use the metric $\alpha(S) = \mathbb{E}_{\boldsymbol{x} \in \boldsymbol{X}}[|I(S|\boldsymbol{x} + \boldsymbol{\delta}) - I(S|\boldsymbol{x})|] / \mathbb{E}_{\boldsymbol{x} \in \boldsymbol{X}}[|I(S|\boldsymbol{x})|]$ to evaluate the sensitivity of the interactive concept $S$ to adversarial perturbations, where $\delta$ denotes the adversarial perturbation generated by the $\ell_\infty$ attack [33]. In this way, a small $\alpha(S)$ value indicates that the interactive concept $S$ is robust to the adversarial attack.

We follow experimental settings in the *experimental verification* paragraph of Section 3.2 to evaluate different DNNs. Figure 6 shows that compared to high-order interactive concepts, low-order concepts usually obtain smaller $A^{(s)} = \mathbb{E}_{S \subseteq N, |S|=s}[\alpha(S)]$ values. Such phenomena demonstrate that low-order interactive concepts are more robust to adversarial attacks.

## 4.2 Connections to existing findings on what are learned first by a DNN

In this subsection, we discuss some related studies on which kind of knowledge is usually first learned by a DNN. Most previous studies conducted experiments to explore the knowledge that was easier to be learned by a DNN, without providing much theoretical support. However, we find that our theorems can partially explain mechanisms behind some previous findings.

• Arpit et al. [5] trained DNNs to classify both normal samples and white-noise samples to different object categories. In this way, they considered that the DNN encoded simple concepts to classify normal samples, but the DNN had to learn complex concepts to classify white noises to randomly-assigned labels. They observed that the DNN usually learned normal samples first, because the classification accuracy of normal samples increased before that of white noises. To this end, our research provides more insights into such an observation. Specifically, Cheng et al. [8] have proven that the classification of noisy data usually depends on high-order concepts, *i.e.*, the classification of noisy data forces the DNN to memorize each specific white-noise sample as a specific high-order interactive concept. Let us combine this conclusion with our finding that high-order interactive concepts are hard to learn. Then, we can easily owe the slow learning of white-noise samples observed in [5] to the difficulty of learning high-order concepts.

• Mangalam and Prabhu [34] considered/defined easy samples as training samples that could be correctly classified by shallow machine learning models, such as support vector machine (SVM) and random forests (RF). They discovered that DNNs first learned easy samples, and then gradually learned more about hard samples. To this end, our research verifies such observation. Specifically, we claim that easy samples mainly contain low-order interactive concepts. Thus, hard samples mainly contain high-order interactive concepts corresponding to complex interactions between numerous

input variables, and thus are difficult to be classified by shallow models (*e.g.*, the SVM and the RF). In this way, the fast learning of low-order concepts is another understanding of the finding in [34].

• Xu et al. [63] discovered that during the training process, DNNs usually first learned samples of low frequencies (*e.g.,* robust to noises), and then encoded samples of high frequencies (*e.g.,* sensitive to noises). However, the original design of DNNs is not towards learning specific spectrums, and techniques of deep learning are not developed by assuming a periodic loss landscape of training samples. Therefore, we believe there should be a more direct explanation for the spectrum-learning phenomenon discovered by [63]. To this end, our research explains this phenomenon as the difficulty of learning high-order concepts. According to Section 4.1, low-order concepts usually are less sensitive to input perturbations, thereby corresponding to low-frequency components defined in the loss landscape in [63]. Accordingly, high-order concepts correspond to high-frequency components.

• Liu et al. [28] discovered that during adversarial training, the training loss of the DNN trained on easy samples decreased faster than that of the DNN trained on hard samples. To this end, we consider easy samples in adversarial training mentioned by [28] may mainly contain low-order interactive concepts. It is because, as discussed in Section 4.1, [43] discovered that low-order interactions were robust to adversarial perturbations. Thus, the fast learning of easy samples in adversarial training can be roughly owing to the learning of low-order concepts.

## 5  Conclusion and discussion

In this paper, we theoretically explain the trend of DNNs learning simple concepts. Since the emergence of interactive concepts of DNNs has been observed [45, 27] and proved [47], we aim to theoretically explain the explicit theoretical connection between the conceptual complexity and the difficulty of learning concepts. In this way, we prove that low-order interactive concepts in the data are much more stable than high-order interactive concepts, which makes low-order interactive concepts more likely to be encoded. Besides, our research can also provide new insights into several previous empirical understandings [5, 34, 63, 28] *w.r.t.* the conceptual representation of DNNs.

There are very few ways to define and examine what is a "concept." In this paper, we only use the following three properties to support the faithfulness of using sparse salient interactions as concepts encoded by the DNN.
• Ren et al. [47] have proved that a well-trained DNN can encode just a small number of salient interactions for inference under some common conditions.
• Ren et al. [45] have proved that, such a small number of salient interactions extracted from a sample can well mimic DNN's outputs on numerous masked samples.
• Li and Zhang [27] have discovered that salient interactions have considerable transferability and strong discrimination power.
However, the above properties cannot guarantee a clear correspondence between an interactive concept and a concept in human cognition. Up to now, we cannot mathematically formulate what is a concept in cognitive science. Thus, there is still a long way to unify the learning difficulty of a DNN and the cognitive difficulty of human beings.

## Acknowledgments

This work is partially supported by the National Nature Science Foundation of China (62276165), National Key R&D Program of China (2021ZD0111602), Shanghai Natural Science Foundation (21JC1403800,21ZR1434600), National Nature Science Foundation of China (U19B2043). This paper is also partially supported by Huawei Technologies Inc.

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

# A  Literature on the representation capacity of DNNs

Formulating and evaluating the representation ability of DNNs is an emerging perspective to explain DNNs. Pascanu et al. [38] and Montufar et al. [35] evaluated the representation capacity of DNNs based on the number of linear response regions. Kornblith et al. [21], Raghu et al. [42], and Morcos et al. [36] analyzed representations similarity between DNNs by using canonical correlation analysis. Shen et al. [52], Chen et al. [7], and Liu et al. [30] measured the quality of knowledge representations encoded in DNNs for point cloud processing. The information-bottleneck theory [54] quantified information encoded in DNNs, and was extended to improve the representation capacity of DNNs [1, 3]. Xu [64] and Xu et al. [63] explained the generalization of DNNs from the perspective of Fourier analysis. Furthermore, several metrics were proposed to evaluate the robustness or generalization capacity of DNNs, including the flatness of loss functions at minima [13], the stability of optimization [17], the CLEVER score [62], the stiffness [14], and the sensitivity metrics [37].

In contrast to previous empirical studies, we mathematically formulate concepts encoded by DNNs and theoretically prove that DNNs mainly learn simple concepts.

# B  Literature on interactions

Many studies investigated interactions between input variables of DNNs in recent years. Grabisch and Roubens [16] proposed the Shapley interaction index to measure the interaction between players in a cooperative game, based on the Shapley value [51]. Lundberg et al. [32] used the Shapley interaction index to analyze tree ensembles. Sundararajan et al. [57] proposed the Shapley-Taylor interaction index, and Tsai et al. [58] defined Faith-Shap, which was another interaction index.

In this paper, we use interactions between input variables of a DNN to represent concepts encoded by the DNN. In this way, We theoretically explain and empirically verify that DNN is easier to learn simple interactive concepts.

# C  Literature on the shortcut learning and simplicity bias of DNNs

Many studies focused on the underlying principles and limitations of DNNs. Geirhos et al. [15], Robinson et al. [48], and Scimeca et al. [49] investigated the *shortcut learning* in DNNs. *I.e.*, DNNs may find decision rules that perform well on standard datasets, but fail to generalize to real-world scenarios. For example, DNNs successfully detected pneumonia by identifying hospital-specific metal tokens on the scan, but they failed to learn much about pneumonia and achieved low performance for scans from novel hospitals. Meanwhile, many studies proposed *simplicity bias*, *i.e.,* standard training procedures for DNNs have a bias towards learning simple models [50, 20, 40]. Specifically, DNNs tend to learn low-rank embeddings [20] or simple features [50, 29, 40, 2].

From the perspective of the simplicity bias, our study considers the definition of interactive concepts in [47], and analyzes the bias towards learning simple concepts. This work clarifies an exact form of the complexity of concepts that a DNN is difficult to learn.

# D  Axioms and theorems for the Harsanyi dividend interaction

In this section, we introduce that the Harsanyi dividend interaction $I(S)$ satisfies several desirable axioms and theorems.

The Harsanyi dividend interactions $I(S)$ satisfies the *efficiency, linearity, dummy, symmetry, anonymity, recursive* and *interaction distribution* axioms, as follows.

(1) *Efficiency axiom* (proved by [18]). The reward of a neural network can be decomposed into interaction effects of different contexts, *i.e.* $v(N) = \sum_{S \subseteq N} I(S)$.

(2) *Linearity axiom*. If we merge rewards of two neural networks $w$ and $v$ as the reward of model $u$, *i.e.* $\forall S \subseteq N, u(S) = w(S) + v(S)$, then their interaction effects $I_v(S)$ and $I_w(S)$ can be represented as $\forall S \subseteq N, I_u(S) = I_v(S) + I_w(S)$.

(3) *Dummy axiom*. If a variable $i \in N$ is a dummy variable, *i.e.* $\forall S \subseteq N \backslash \{i\}, v(S \cup \{i\}) = v(S) + v(\{i\})$, then it does not interact with other variables, $\forall \emptyset \neq S \subset N \setminus \{i\}, I(S \cup \{i\}) = 0$.

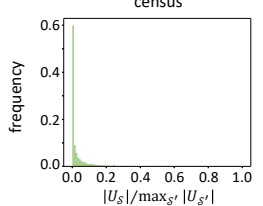 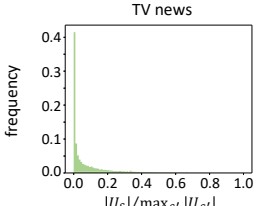

Figure 7: The average histograms of absolute effects of interactive concepts encoded by five-layer MLPs.

(4) *Symmetry axiom.* Given two input variables $i, j \in N$, if $\forall S \subseteq N\backslash\{i,j\}, v(S \cup \{i\}) = v(S \cup \{j\})$, then $\forall S \subseteq N\backslash\{i,j\}, I(S \cup \{i\}) = I(S \cup \{j\})$.

(5) *Anonymity axiom.* For any permutations $\pi$ on $N$, we have $\forall S \subseteq N, I_v(S) = I_{\pi v}(\pi S)$, where $\pi S \{\pi(i)|i \in S\}$, and the new model $\pi v$ is defined by $(\pi v)(\pi S) = v(S)$. This indicates that interaction effects are not changed by permutation.

(6) *Recursive axiom.* The interaction effects can be calculated recursively. Given $i \in N$ and $S \subseteq N\backslash\{i\}$, the interaction effect of the pattern $S\cup\{i\}$ can be represented as the interaction effect of $S$ with $i$ minus the interaction effect of $S$ without $i$, *i.e.* $\forall S \subseteq N\backslash\{i\}, I(S \cup \{i\}) = I(S|i$ is always present$) - I(S)$. $I(S|i$ is always present$)$ denotes the interaction effect when the variable $i$ is always present as a constant context, *i.e.* $I(S|i$ is always present$) = \sum_{L \subseteq S}(-1)^{|S|-|L|} \cdot v(L \cup \{i\})$.

(7) *Interaction distribution axiom.* This axiom shows how interactions are distributed for "interaction functions" [57]. The definition of the interaction function $v_T$ parameterized by a subset of variables $T$ is given as follows. $\forall S \subseteq N$, if $T \subseteq S, v_T(S) = c$; otherwise, $v_T(S) = 0$. The function $v_T$ models pure interaction among the variables in $T$, because only if all variables in $T$ are present, the output value will be increased by $c$. The interactions encoded in the function $v_T$ satisfies $I(T) = c$, and $\forall S \neq T, I(S) = 0$.

# E Sparsity of interactive concepts

In this section, we conducted experiments to verify the sparsity of interactive concepts. To this end, we computed effects $U_S$ of all $2^n$ interactive concepts encoded by a DNN following [45]. Specifically, we trained a five-layer MLP on the *census* dataset and the *TV news* dataset), respectively. For better visualization, we re-scaled effects of interactive concepts $U_S$ by $|U_S|/\max_{S' \subseteq N}|U_{S'}|$. Moreover, the strength of effects are average over different samples in each dataset. Fig. 7 shows histograms of absolute effects of interactive concepts.

# F Complexity of interactive concepts

Many previous studies [11, 67, 59, 45, 43] used multi-order interactions to analyze DNNs. Specifically, Given a pre-trained DNN $v$ and a masked sample $\boldsymbol{x}_S$, the multi-order interaction $I^{(m)}(i,j)$ used in [59, 11, 67] is given as follows:

$$I^{(m)}(i,j) = \mathbb{E}_{S \subseteq N\backslash\{i,j\}, |S|=m}[\Delta v(i,j,S)], \tag{1}$$

where $\Delta v(i,j,S) = v(\boldsymbol{x}_{S\cup\{i,j\}}) - v(\boldsymbol{x}_{S\cup\{i\}}) - v(\boldsymbol{x}_{S\cup\{j\}}) + v(\boldsymbol{x}_S)$. They consider that $I^{(m)}(i,j)$ reflects the collaboration with $m$ contextual variables ($m$ pixels). In this way, the order $m$ measures the number of variables in $S$. Therefore, a low-order interaction denotes a relatively simple collaboration between input variables with a small context $S$. In contrast, a high-order interaction represents a complex collaboration between input variables with a large context $S$.

In this paper, we consider that the number of variables in an interactive concept $S$ can measure the complexity of an interactive concept encoded by a DNN, namely *the order of the interactive concept*, complexity$(S)$=order$(S)$=$|S|$. Thus, low-order concepts usually represent simple AND relationships among a few input variables. In comparison, high-order concepts often refer to as relatively complex AND relationships among a large number of input variables.

# G   Proof of Theorems

## G.1   Proof of Theorem 2 in the main paper

**Theorem 2.** *Given a neural network $v$ and an arbitrary input sample $\boldsymbol{x}' \in \mathbb{R}^n$, the network output can be decomposed by using the Taylor expansion [10], i.e., $v(\boldsymbol{x}') = \sum_{S \subseteq N} \sum_{\boldsymbol{\pi} \in Q_S} U_{S,\boldsymbol{\pi}} \cdot J(S, \boldsymbol{\pi}|\boldsymbol{x}')$. In this way, according to Eq. (1), the interaction effect $I(S|\boldsymbol{x}')$ on the sample $\boldsymbol{x}'$ can be reformulated as follows.*

$$I(S|\boldsymbol{x}') = \sum_{\boldsymbol{\pi} \in Q_S} U_{S,\boldsymbol{\pi}} \cdot J(S, \boldsymbol{\pi}|\boldsymbol{x}'). \tag{2}$$

*Here, $J(S, \boldsymbol{\pi}|\boldsymbol{x}') = \prod_{i \in S} \left( \text{sign}(x_i' - b_i) \cdot \frac{x_i' - b_i}{\tau} \right)^{\pi_i}$ denotes a Taylor expansion term of the degree $\boldsymbol{\pi}$, where the degree $\boldsymbol{\pi} \in Q_S = \{[\pi_1, \ldots, \pi_n] | \forall i \in S, \pi_i \in \mathbb{N}^+; \forall i \notin S, \pi_i = 0\}$ and $b_i$ is the baseline value to mask the input variable $x_i$. In addition, $U_{S,\boldsymbol{\pi}} = \frac{\tau^m}{\prod_{i=1}^n \pi_i!} \frac{\partial^m v(\boldsymbol{x}'_\emptyset)}{\partial x_1^{\pi_1} \cdots \partial x_n^{\pi_n}} \cdot \prod_{i \in S} [\text{sign}(x_i' - b_i)]^{\pi_i}$, where $\boldsymbol{x}'_\emptyset$ denotes the sample whose input variables are all masked. $m = \sum_{i=1}^n \pi_i$.*

*Proof.* Let us denote the function on the right of Eq. (2) by $\tilde{I}(S|\boldsymbol{x}')$, *i.e.*,

$$\tilde{I}(S|\boldsymbol{x}') = \sum_{\pi \in Q_S} U_{S,\pi} J(S, \pi|\boldsymbol{x}') \tag{3}$$

We need to prove that for any arbitrary input sample $\forall \boldsymbol{x}' \in \mathbb{R}^n$, $\tilde{I}(S|\boldsymbol{x}') = I(S|\boldsymbol{x}')$.

Actually, it has been proven in [16] and [45] that the Harsanyi dividend $I(S|\boldsymbol{x}')$ is the **unique** metric satisfying the faithfulness requirement mentioned in the main paper, *i.e.*, satisfying

$$\forall\, T \subseteq N,\ v(\boldsymbol{x}'_T) = \sum_{S \in \Omega, S \subseteq T} I(S|\boldsymbol{x}'). \tag{4}$$

Thus, as long as we can prove that $\tilde{I}(S|\boldsymbol{x}')$ also satisfies the above faithfulness requirement, we can obtain $\tilde{I}(S|\boldsymbol{x}') = I(S|\boldsymbol{x}')$.

To this end, we only need to prove $\tilde{I}(S|\boldsymbol{x}')$ also satisfies the faithfulness requirement in Eq. (4). Specifically, given an input sample $\forall \boldsymbol{x}' \in \mathbb{R}^n$, let us consider the Taylor expansion of the network output $v(\boldsymbol{x}_T)$ of an arbitrarily masked sample $\boldsymbol{x}'_T (T \subseteq N)$, which is expanded at $\boldsymbol{x}'_\emptyset = [b_1, \ldots, b_n]$. Then, we have

$$\forall\, T \subseteq N, \quad v(\boldsymbol{x}'_T) = \sum_{\pi_1 = 0}^\infty \sum_{\pi_2 = 0}^\infty \cdots \sum_{\pi_n = 0}^\infty \frac{1}{\prod_{i=1}^n \pi_i!} \frac{\partial^m v(\boldsymbol{x}'_\emptyset)}{\partial x_1^{\pi_1} \cdots \partial x_n^{\pi_n}} \cdot \prod_{i=1}^n [(\boldsymbol{x}'_T)_i - b_i]^{\pi_i}, \tag{5}$$

where $\boldsymbol{\pi} \in \{[\pi_1, \ldots, \pi_n] | \forall i \in N, \pi_i \in \mathbb{N}\}$ denotes the degree vector of Taylor expansion terms, and $m = \sum_{i=1}^n \pi_i$. In addition, $b_i$ denotes the reference value to mask the input variable $x_i$.

According to the definition of the masked sample $\boldsymbol{x}'_T$, we have that all variables in $T$ keep unchanged and other variables are masked to the reference value. That is, $\forall\, i \in T, (\boldsymbol{x}'_T)_i = x_i; \forall\, i \notin T, (\boldsymbol{x}'_T)_i = b_i$. Hence, we obtain $\forall i \notin T, [(\boldsymbol{x}'_T)_i - b_i]^{\pi_i} = 0$. Then, among all Taylor expansion terms, only terms corresponding to degrees $\boldsymbol{\pi}$ in the set $P = \{[\pi_1, \ldots, \pi_n] | \forall i \in T, \pi_i \in \mathbb{N}; \forall i \notin T, \pi_i = 0\}$ may not be zero. Therefore, Eq. (5) can be rewritten as

$$\forall\, T \subseteq N, \quad v(\boldsymbol{x}'_T) = \sum_{\boldsymbol{\pi} \in P} \frac{1}{\prod_{i=1}^n \pi_i!} \frac{\partial^m v(\boldsymbol{x}'_\emptyset)}{\partial x_1^{\pi_1} \cdots \partial x_n^{\pi_n}} \cdot \prod_{i \in T} (x_i' - b_i)^{\pi_i}. \tag{6}$$

We find that the set $P$ can be divided into multiple disjoint sets as follows, $P = \cup_{S \subseteq T} Q_S$, where $Q_S = \{[\pi_1, \ldots, \pi_n] | \forall i \in S, \pi_i \in \mathbb{N}^+; \forall i \notin S, \pi_i = 0\}$. Then, we can derive that

$$\forall\, T \subseteq N, \quad v(\boldsymbol{x}'_T) = \sum_{S \subseteq T} \sum_{\boldsymbol{\pi} \in Q_S} \frac{1}{\prod_{i=1}^n \pi_i!} \frac{\partial^m v(\boldsymbol{x}'_\emptyset)}{\partial x_1^{\pi_1} \cdots \partial x_n^{\pi_n}} \cdot \prod_{i \in S} (x_i' - b_i)^{\pi_i}$$

$$= \sum_{S \subseteq T} \sum_{\boldsymbol{\pi} \in Q_S} \underbrace{\frac{\tau^m}{\prod_{i=1}^n \pi_i!} \frac{\partial^m v(\boldsymbol{x}'_\emptyset)}{\partial x_1^{\pi_1} \cdots \partial x_n^{\pi_n}} \prod_{i \in S} (\delta_i)^{\pi_i}}_{\text{termed } U_{S,\pi}} \cdot \underbrace{\prod_{i \in S} (\delta_i \frac{x_i' - b_i}{\tau})^{\pi_i}}_{\text{termed } J(S, \pi|\boldsymbol{x}')}, \tag{7}$$

where $\tau \in \mathbb{R}$ is a pre-defined constant and $\delta_i = \text{sign}(x_i - r_i) \in \{-1, 1\}$ is a sign function satisfying $\delta_i^{2m} = 1$ ($m = \sum_{i=1}^n \pi_i$). Then, Eq. (7) can be re-written as

$$\forall\, T \subseteq N,\ v(\boldsymbol{x}'_T) = \sum_{S \subseteq T} \sum_{\boldsymbol{\pi} \in Q_S} U_{S,\boldsymbol{\pi}} \cdot J(S, \boldsymbol{\pi}|\boldsymbol{x}') = \sum_{S \subseteq T} \tilde{I}(S|\boldsymbol{x}'). \tag{8}$$

Thus, $\tilde{I}(S|\boldsymbol{x}')$ satisfies the faithfulness requirement in Eq. (4) when $\Omega = 2^N$.

Therefore, Theorem 2 holds. $\qquad\qquad\qquad\qquad\qquad\qquad\qquad\qquad\qquad\qquad\qquad\qquad$ $\square$

## G.2 Proof of Theorem 3 in the main paper

**Theorem 3.** *Let us add a Gaussian perturbation $\epsilon \sim \mathcal{N}(\mathbf{0}, \sigma^2 \boldsymbol{I})$ to the input sample $\boldsymbol{x}$. Let us first consider the case with the lowest degree $\hat{\boldsymbol{\pi}} = [\hat{\pi}_1, \ldots, \hat{\pi}_n] \in Q_S$, satisfying that $\forall i \in S, \hat{\pi}_i = 1; \forall i \notin S, \hat{\pi}_i = 0$. The mean and variance of $J(S, \hat{\boldsymbol{\pi}}|\boldsymbol{x} + \epsilon)$ over the Gaussian perturbation $\epsilon$ are given as*

$$\mathbb{E}_{\epsilon}[I(S|\boldsymbol{x} + \epsilon)] = U_{S,\hat{\boldsymbol{\pi}}}, \quad \mathrm{Var}_{\epsilon}[I(S|\boldsymbol{x} + \epsilon)] = U_{S,\hat{\boldsymbol{\pi}}}^2 \left[ \left(1 + (\sigma/\tau)^2\right)^s - 1 \right].$$

*Furthermore, for the more general case with an arbitrary degree $\boldsymbol{\pi} \in Q_S = \{[\pi_1, \cdots, \pi_n] | \forall i \in S, \pi_i \in \mathbb{N}^+; \forall i \notin S, \pi_i = 0\}$, the mean and variance of $J(S, \boldsymbol{\pi}|\boldsymbol{x} + \epsilon)$ are computed as*

$$\mathbb{E}_{\epsilon}[J(S, \boldsymbol{\pi}|\boldsymbol{x} + \epsilon)] = \mathbb{E}_{\epsilon}[\textstyle\prod_{i \in S}(1 + \epsilon_i/\tau)^{\pi_i}], \quad \mathrm{Var}_{\epsilon}[J(S, \boldsymbol{\pi}|\boldsymbol{x} + \epsilon)] = \mathrm{Var}_{\epsilon}[\textstyle\prod_{i \in S}(1 + \epsilon_i/\tau)^{\pi_i}].$$

*Proof.* If we only consider Taylor expansion term of the lowest degree, then $I(S|\boldsymbol{x}') \approx U_{S,\hat{\boldsymbol{\pi}}} \cdot J(S, \hat{\boldsymbol{\pi}}|\boldsymbol{x}')$, where $J(S, \hat{\boldsymbol{\pi}}|\boldsymbol{x}') = \prod_{i \in S} \mathrm{sign}(x_i' - b_i) \cdot \frac{x_i' - b_i}{\tau}$.

Let us add a Gaussian perturbation $\epsilon \sim \mathcal{N}(\mathbf{0}, \sigma^2 \boldsymbol{I})$ to the input sample $\boldsymbol{x}$. Then, we have

$$
\begin{aligned}
I(S|\boldsymbol{x} + \epsilon) &\approx U_{S,\hat{\boldsymbol{\pi}}} \cdot J(S, \hat{\boldsymbol{\pi}}|\boldsymbol{x} + \epsilon) \\
J(S, \hat{\boldsymbol{\pi}}|\boldsymbol{x} + \epsilon) &= \prod_{i \in S} \mathrm{sign}(x_i + \epsilon_i - b_i) \cdot \frac{x_i + \epsilon_i - b_i}{\tau} \\
&= \prod_{i \in S} \left( \mathrm{sign}(x_i + \epsilon_i - b_i) \cdot \frac{x_i - b_i}{\tau} + \mathrm{sign}(x_i + \epsilon_i - b_i) \cdot \frac{\epsilon_i}{\tau} \right)
\end{aligned}
\tag{9}
$$

According to the setting of the baseline value, we have $\forall i \in S, x_i - b_i \in \{-\tau, \tau\}$. In Section 2.2, we have assumed that the perturbation is small, *i.e.*, $\forall i \in S, |\epsilon_i| \leq \tau$. In this way, we have $\mathrm{sign}(x_i + \epsilon_i - b_i) = \mathrm{sign}(x_i - b_i)$, and we can obtain

$$
\begin{aligned}
J(S, \hat{\boldsymbol{\pi}}|\boldsymbol{x} + \epsilon) &= \prod_{i \in S} \left( \mathrm{sign}(x_i - b_i) \cdot \frac{x_i - b_i}{\tau} + \mathrm{sign}(x_i - b_i) \cdot \frac{\epsilon_i}{\tau} \right) \\
&= \prod_{i \in S} \left( 1 + \mathrm{sign}(x_i - b_i) \cdot \frac{\epsilon_i}{\tau} \right)
\end{aligned}
\tag{10}
$$

$$
\begin{aligned}
\Rightarrow \mathbb{E}_{\epsilon}[J(S, \hat{\boldsymbol{\pi}}|\boldsymbol{x} + \epsilon)] &= \mathbb{E}_{\epsilon}\left[ \prod_{i \in S} \left( 1 + \mathrm{sign}(x_i - b_i) \cdot \frac{\epsilon_i}{\tau} \right) \right] \\
\mathrm{Var}_{\epsilon}[J(S, \hat{\boldsymbol{\pi}}|\boldsymbol{x} + \epsilon)] &= \mathrm{Var}_{\epsilon}\left[ \prod_{i \in S} \left( 1 + \mathrm{sign}(x_i - b_i) \cdot \frac{\epsilon_i}{\tau} \right) \right]
\end{aligned}
\tag{11}
$$

Since $\mathrm{sign}(x_i - b_i) \in \{-1, 1\}$, we have $1 + \mathrm{sign}(x_i - b_i) \cdot \frac{\epsilon_i}{\tau} \sim \mathcal{N}(1, (\sigma/\tau)^2), \forall i \in S$.

**Proposition 1.** *If random variables $X_1, X_2, \cdots, X_k$ are independent of each other, then $\mathbb{E}[X_1 X_2 \cdots X_k] = \prod_{i=1}^{k} \mathbb{E}[X_i]$, and $\mathrm{Var}[X_1 X_2 \cdots X_k] = \prod_{i=1}^{k}(\mathbb{E}[X_i]^2 + \mathrm{Var}[X_i]^2) - \prod_{i=1}^{k} \mathbb{E}[X_i]^2$.*

According to the above proposition, we have

$$
\begin{aligned}
\mathbb{E}_{\epsilon}[J(S, \hat{\boldsymbol{\pi}}|\boldsymbol{x} + \epsilon)] &= \prod_{i \in S} 1 = 1 \\
\mathrm{Var}_{\epsilon}[J(S, \hat{\boldsymbol{\pi}}|\boldsymbol{x} + \epsilon)] &= \prod_{i \in S} \left( 1^2 + (\sigma/\tau)^2 \right) - \prod_{i \in S} 1^2 \\
&= \left( 1 + (\sigma/\tau)^2 \right)^{|S|} - 1
\end{aligned}
\tag{12}
$$

Therefore,

$$\mathbb{E}_{\boldsymbol{\epsilon}}[I(S|\boldsymbol{x}+\boldsymbol{\epsilon})] = \mathbb{E}_{\boldsymbol{\epsilon}}[U_{S,\hat{\boldsymbol{\pi}}} \cdot J(S,\hat{\boldsymbol{\pi}}|\boldsymbol{x}+\boldsymbol{\epsilon})] = U_{S,\hat{\boldsymbol{\pi}}}$$

$$\text{Var}_{\boldsymbol{\epsilon}}[I(S|\boldsymbol{x}+\boldsymbol{\epsilon})] = \text{Var}_{\boldsymbol{\epsilon}}[U_{S,\hat{\boldsymbol{\pi}}} \cdot J(S,\hat{\boldsymbol{\pi}}|\boldsymbol{x}+\boldsymbol{\epsilon})] = U_{S,\hat{\boldsymbol{\pi}}}^2 \left( \left(1+(\sigma/\tau)^2\right)^{|S|} - 1 \right) \tag{13}$$

According to Theorem 2, given an arbitrary input sample $\boldsymbol{x}'$, we have

$$J(S,\boldsymbol{\pi}|\boldsymbol{x}') = \prod_{i \in S} \left( \text{sign}(x_i' - b_i) \cdot \frac{x_i' - b_i}{\tau} \right)^{\pi_i} \tag{14}$$

Let us add a Gaussian perturbation $\boldsymbol{\epsilon} \sim \mathcal{N}(\boldsymbol{0}, \sigma^2 \boldsymbol{I})$ to the input sample $\boldsymbol{x}$. In this way, we have

$$
\begin{aligned}
J(S,\boldsymbol{\pi}|\boldsymbol{x}+\boldsymbol{\epsilon}) &= \prod_{i \in S} \left( \text{sign}(x_i + \epsilon_i - b_i) \cdot \frac{x_i + \epsilon_i - b_i}{\tau} \right)^{\pi_i} \\
&= \prod_{i \in S} \left( \text{sign}(x_i + \epsilon_i - b_i) \cdot \frac{x_i - b_i}{\tau} + \text{sign}(x_i + \epsilon_i - b_i) \cdot \frac{\epsilon_i}{\tau} \right)^{\pi_i}
\end{aligned} \tag{15}
$$

According to the setting of the baseline value, $\forall i \in S, x_i - b_i \in \{-\tau, \tau\}$. We assume that the perturbation is small, *i.e.*, $\forall i \in S, |\epsilon_i| \ll \tau$. In this way, $\text{sign}(x_i + \epsilon_i - b_i) = \text{sign}(x_i - b_i)$, and we can obtain

$$
\begin{aligned}
J(S,\boldsymbol{\pi}|\boldsymbol{x}+\boldsymbol{\epsilon}) &= \prod_{i \in S} \left( \text{sign}(x_i - b_i) \cdot \frac{x_i - b_i}{\tau} + \text{sign}(x_i - b_i) \cdot \frac{\epsilon_i}{\tau} \right)^{\pi_i} \\
&= \prod_{i \in S} \left( 1 + \text{sign}(x_i - b_i) \cdot \frac{\epsilon_i}{\tau} \right)^{\pi_i}
\end{aligned} \tag{16}
$$

$$
\begin{aligned}
\Rightarrow \mathbb{E}_{\boldsymbol{\epsilon}}[J(S,\boldsymbol{\pi}|\boldsymbol{x}+\boldsymbol{\epsilon})] &= \mathbb{E}_{\boldsymbol{\epsilon}} \left[ \prod_{i \in S} \left( 1 + \text{sign}(x_i - b_i) \cdot \frac{\epsilon_i}{\tau} \right)^{\pi_i} \right] \\
\text{Var}_{\boldsymbol{\epsilon}}[J(S,\boldsymbol{\pi}|\boldsymbol{x}+\boldsymbol{\epsilon})] &= \text{Var}_{\boldsymbol{\epsilon}} \left[ \prod_{i \in S} \left( 1 + \text{sign}(x_i - b_i) \cdot \frac{\epsilon_i}{\tau} \right)^{\pi_i} \right]
\end{aligned} \tag{17}
$$

Since $\forall i \in S, \epsilon_i$ is independent of each other, according to Proposition 1 and Eq. (17), we have

$$\mathbb{E}_{\boldsymbol{\epsilon}}[J(S,\boldsymbol{\pi}|\boldsymbol{x}+\boldsymbol{\epsilon})] = \prod_{i \in S} \mathbb{E}_{\epsilon_i} \left[ \left( 1 + \text{sign}(x_i - b_i) \cdot \frac{\epsilon_i}{\tau} \right)^{\pi_i} \right]$$

$$\text{Var}_{\boldsymbol{\epsilon}}[J(S,\boldsymbol{\pi}|\boldsymbol{x}+\boldsymbol{\epsilon})] = \prod_{i \in S} \mathbb{E}_{\epsilon_i} \left[ \left( 1 + \text{sign}(x_i - b_i) \cdot \frac{\epsilon_i}{\tau} \right)^{2\pi_i} \right] - \prod_{i \in S} \left( \mathbb{E}_{\epsilon_i} \left[ \left( 1 + \text{sign}(x_i - b_i) \cdot \frac{\epsilon_i}{\tau} \right)^{\pi_i} \right] \right)^2 \tag{18}$$

Since $\text{sign}(x_i - b_i) \in \{-1, 1\}$, we have $\mathbb{E}_{\epsilon_i} \left[ \left( 1 + \text{sign}(x_i - b_i) \cdot \frac{\epsilon_i}{\tau} \right)^k \right] = \mathbb{E}_{\epsilon_i} \left[ \left( 1 + \frac{\epsilon_i}{\tau} \right)^k \right], \forall k \in \mathbb{N}^+$. Therefore, we obtain

$$\mathbb{E}_{\boldsymbol{\epsilon}}[J(S,\boldsymbol{\pi}|\boldsymbol{x}+\boldsymbol{\epsilon})] = \prod_{i \in S} \mathbb{E}_{\epsilon_i} \left[ \left( 1 + \frac{\epsilon_i}{\tau} \right)^{\pi_i} \right]$$

$$= \mathbb{E}_{\boldsymbol{\epsilon}} \left[ \prod_{i \in S} \left( 1 + \frac{\epsilon_i}{\tau} \right)^{\pi_i} \right]$$

$$\text{Var}_{\boldsymbol{\epsilon}}[J(S,\boldsymbol{\pi}|\boldsymbol{x}+\boldsymbol{\epsilon})] = \prod_{i \in S} \mathbb{E}_{\epsilon_i} \left[ \left( 1 + \frac{\epsilon_i}{\tau} \right)^{2\pi_i} \right] - \prod_{i \in S} \left( \mathbb{E}_{\epsilon_i} \left[ \left( 1 + \frac{\epsilon_i}{\tau} \right)^{\pi_i} \right] \right)^2$$

$$= \text{Var}_{\boldsymbol{\epsilon}} \left[ \prod_{i \in S} \left( 1 + \frac{\epsilon_i}{\tau} \right)^{\pi_i} \right].$$

$\square$

## G.3 Proof of Theorem 4 in the main paper

**Theorem 4.** *Given an arbitrarily masked sample $\boldsymbol{x}_T (\forall T \subseteq N)$, the function $C_S(\boldsymbol{x}_T)$ defined above can well fit the binary activation state of the concept $S$ in the sample $\boldsymbol{x}_T$.*

$$\forall T \subseteq N, \; C(S|\boldsymbol{x}_T) = \prod\nolimits_{i \in S} A_i(\boldsymbol{x}_T) = \mathbb{1}(S \subseteq T), \tag{19}$$

*where $A_i(\boldsymbol{x}_T) \in \{0,1\}$ denotes whether the variable $x_i$ is present $A_i(\boldsymbol{x}_T) = 1$ or being masked $A_i(\boldsymbol{x}_T) = 0$ in the sample $\boldsymbol{x}_T$.*

*Proof.* We know that the function of the activation state on an arbitrary sample $\boldsymbol{x}'$ is given by

$$C_S(\boldsymbol{x}') = \sum\nolimits_{\boldsymbol{\pi} \in Q_S} U_{S,\boldsymbol{\pi}} J(S, \boldsymbol{\pi}|\boldsymbol{x}')/U_S,$$
$$\text{where } J(S, \boldsymbol{\pi}|\boldsymbol{x}') = \prod\nolimits_{i \in S} \left( \text{sign}(x_i' - b_i) \cdot \frac{x_i' - b_i}{\tau} \right)^{\pi_i} \tag{20}$$

Specifically, now we consider a masked sample $\boldsymbol{x}_T$, and we will prove that $C_S(\boldsymbol{x}_T) = \mathbb{1}(S \subseteq T)$. We consider the following two cases.

**Case 1: $S \subsetneq T$.** Then there exists some $j \in S \setminus T$. Since $j \notin T$, according to the masking rule of the sample $\boldsymbol{x}_T$, we have $(\boldsymbol{x}_T)_j - b_j = 0$. Since $j \in S$, we have

$$\forall \pi \in Q_S, \quad J(S, \boldsymbol{\pi}|\boldsymbol{x}_T) = \prod\nolimits_{i \in S} \left( \text{sign}((\boldsymbol{x}_T)_i - r_i) \cdot \frac{(\boldsymbol{x}_T)_i - b_i}{\tau} \right)^{\pi_i} = 0 \tag{21}$$

In this way, we have $C_S(\boldsymbol{x}_T) = \sum_{\boldsymbol{\pi} \in Q_S} U_{S,\boldsymbol{\pi}} J(S, \boldsymbol{\pi}|\boldsymbol{x}_T)/U_S = 0$.

**Case 2: $S \subseteq T$.** In this case, $\forall i \in S$, we have $i \in T$. According to the setting of the reference value in Section 2.3 of the main paper, we have $(\boldsymbol{x}_T)_i - b_i \in \{\tau, -\tau\}$. Then we have $\text{sign}((\boldsymbol{x}_T)_i - b_i) \cdot \frac{(\boldsymbol{x}_T)_i - b_i}{\tau} = 1, \forall i \in S$. This further implies that

$$\forall \pi \in Q_S, \quad J(S, \boldsymbol{\pi}|\boldsymbol{x}_T) = \prod\nolimits_{i \in S} \left( \text{sign}((\boldsymbol{x}_T)_i - b_i) \cdot \frac{(\boldsymbol{x}_T)_i - b_i}{\tau} \right)^{\pi_i} = 1. \tag{22}$$

Therefore, we can derive

$$C_S(\boldsymbol{x}_T) = \sum\nolimits_{\boldsymbol{\pi} \in Q_S} U_{S,\boldsymbol{\pi}}/U_S. \tag{23}$$

Recall that $U_S = I(S|\boldsymbol{x}) = \sum_{\boldsymbol{\pi} \in Q_S} U_{S,\boldsymbol{\pi}} J(S, \boldsymbol{\pi}|\boldsymbol{x})$, in which $J(S, \boldsymbol{\pi}|\boldsymbol{x}) = \prod_{i \in S} \left( \text{sign}(x_i - r_i) \cdot \frac{x_i - b_i}{\tau} \right)^{\pi_i} = 1$, because $x_i - b_i \in \{\tau, -\tau\}$. So actually, we have

$$U_S = \sum\nolimits_{\boldsymbol{\pi} \in Q_S} U_{S,\boldsymbol{\pi}}. \tag{24}$$

Therefore, $C_S(\boldsymbol{x}_T) = \sum_{\boldsymbol{\pi} \in Q_S} U_{S,\boldsymbol{\pi}}/U_S = 1$.

Combining the two cases, we can conclude that $C_S(\boldsymbol{x}_T) = \mathbb{1}(S \subseteq T) = \prod_{i \in S} A_i(\boldsymbol{x}_T)$.

$\square$

## G.4 Relation between interactive concepts and multi-order interactions

In this section, we derive that high-order interactive concepts (computed via the Harsanyi dividend [18]) can be considered as elementary components for high-order interactions used in [43].

Given a pre-trained DNN $v$ and a masked sample $\boldsymbol{x}_S$, the multi-order interaction $I^{(m)}(i,j)$ used in [43] is given as follows:

$$I^{(m)}(i,j) = \mathbb{E}_{S \subseteq N \setminus \{i,j\}, |S|=m}[\Delta v(i,j,S)], \tag{25}$$

where $\Delta v(i,j,S) = v(\boldsymbol{x}_{S \cup \{i,j\}}) - v(\boldsymbol{x}_{S \cup \{i\}}) - v(\boldsymbol{x}_{S \cup \{j\}}) + v(\boldsymbol{x}_S)$.

Let $\Delta v_T(S) = \sum_{L \subseteq T} (-1)^{|T|-|L|} v(\boldsymbol{x}_{L \cup S})$ denote the marginal benefit of variables in $T \subseteq N \setminus S$, given the environment $S$. In this way, $\Delta v_T(S)$ can be represented as the sum of interaction effects inside $T$ and sub-environments of $S$, *i.e.* $\Delta v_T(S) = \sum_{S' \subseteq S} I(T \cup S')$ [45].

Thus, the $I^{(m)}(i,j)$ can be represented as follows,

$$
\begin{aligned}
I^{(m)}(i,j) &= \mathbb{E}_{S\subseteq N\setminus\{i,j\},|S|=m}[\Delta v(i,j,S)] \\
&= \mathbb{E}_{S\subseteq N\setminus\{i,j\},|S|=m}\Big[\sum_{L\subseteq S} I(L\cup\{i,j\})\Big] \\
&= \frac{1}{\binom{n-2}{m}} \sum_{\substack{S\subseteq N\setminus\{i,j\}\\|S|=m}} \Big[\sum_{L\subseteq S} I(L\cup\{i,j\})\Big] \\
&= \sum_{\substack{L\subseteq N\setminus\{i,j\}\\|L|\le m}} I(L\cup\{i,j\}) \sum_{\substack{L\subseteq S\subseteq N\setminus\{i,j\}\\|S|=m}} \frac{1}{\binom{n-2}{m}} \\
&= \sum_{\substack{L\subseteq N\setminus\{i,j\}\\|L|\le m}} I(L\cup\{i,j\}) \frac{\binom{n-2}{m-l}}{\binom{n-2}{m}} \\
&= \sum_{l=0}^{m} \sum_{\substack{L\subseteq N\setminus\{i,j\}\\|L|=m}} \frac{\binom{n-2}{m-l}}{\binom{n-2}{m}} I(L\cup\{i,j\}).
\end{aligned}
$$

Therefore, we prove that $I^{(m)}(i,j) = \sum_{l=0}^{m} \sum_{\substack{L\subseteq N\setminus\{i,j\}\\|L|=m}} \frac{\binom{n-2}{m-l}}{\binom{n-2}{m}} I(L\cup\{i,j\})$.

### G.5 Theoretical connection between variance and learning difficulty

There is a close connection between the variance of interactive concepts and the learning difficulty. In fact, a similar connection has been discovered by [46]. The theoretical connection is proved as follows. Specifically, we simplify the learning of a DNN into a linear regression problem. In this very simple setting, we assume each $i$-th concept encoded by the DNN to be a specific feature, and use $f_i$ to represent the triggering (presence) state of the $i$-th concept in a training sample. Because according to Eq. (9) in the main paper, the network output $v(\cdot)$ is the sum of a small number of salient interactive concepts, we roughly simplify and rewrite the inference output of the DNN as the following linear function

$$
y = \boldsymbol{w}^\top \boldsymbol{f} = w_1 f_1 + \cdots + w_d f_d \ ,
$$

where $\boldsymbol{f} = [f_1, \cdots, f_d]^\top$. Then, $w_i$ can be viewed as the strength of the DNN encoding the $i$-th interactive concept. In this way, if $w_i \approx 0$, the DNN does not learn the $i$-th interactive concept.

We suppose that training samples are sampled from $f \sim P(f) = N(\mu, \Sigma^2)$ and $y^*$ denotes the ground truth label. Specifically, $\boldsymbol{\mu} = [\mu_1, \mu_2, ..., \mu_d]^\top$ and $\boldsymbol{\Sigma} = \mathrm{diag}(\sigma_1^2, \sigma_2^2, ..., \sigma_d^2)$.

The toy regression problem can be formulated as follows,

$$
L = E_{P(f)}[\frac{1}{2}(\boldsymbol{w}^\top \boldsymbol{f} - y^*)^2] \tag{26}
$$

We derive the optimal weights to the above regression task in three steps.

**Step 1.** The optimal weight $w_i$ for the regression problem in Eq. (26) is computed by

$$
\forall 1 \le i \le d, \quad w_i = \frac{1}{\det \boldsymbol{K}} \det(\boldsymbol{K}_1, \cdots, \boldsymbol{K}_{i-1}, \boldsymbol{\zeta}, \boldsymbol{K}_{i+1}, \cdots, \boldsymbol{K}_d) \tag{27}
$$

where $\boldsymbol{K} = \boldsymbol{\mu}\boldsymbol{\mu}^\top + \boldsymbol{\Sigma^2}$, $\boldsymbol{\zeta} = y^*\boldsymbol{\mu}$, and $\boldsymbol{K}_j$ denotes the $j$-th column of the matrix $\boldsymbol{K}$.

**Step 2.** We further prove that for the optimal $\boldsymbol{w}$, we have

$$
\forall 1 \le i, j \le d, \quad \frac{|w_i|}{|w_j|} = \frac{|\mu_i/\sigma_i^2|}{|\mu_j/\sigma_j^2|} \tag{28}
$$

**Step 3.** Based on Step 2, we can derive $|w_i| \propto |1/\sigma_i^2|$, where $w_i$ denotes the strength of the DNN encoding the $i$-th interactive concept. $|w_i| \propto |1/\sigma_i^2|$ indicates that the learning effect of the $i$-th interactive concept is inversely proportional to the variance of the $i$-th interactive concept. Therefore, interactive concepts with high variances (high-order interactive concepts) are more difficult to learn.

# H   Experimental Settings

**Training details.** We trained AlexNet [23], VGG-11 [55], ResNet-18/20 [19] on the CIFAR-10 dataset [22] and the Tiny ImageNet dataset [25], respectively. We also trained a five-layer MLP on the UCI census dataset (namely *census dataset*) and the UCI TV news dataset (namely *TV news dataset*) [6], respectively. Each layer of the MLP contained 100 neurons. We trained each neural networks for 200 epochs with the SGD optimizer.

**Sampling details.** Since, the computational cost of $I(S|\boldsymbol{x})$ was intolerable in real implementation, we applied the sampling-based approximation method in [67] to calculate $I(S|\boldsymbol{x})$. Due to the high dimension of image data (*e.g.* $224 \times 224$ for ImageNet), we uniformly split the input image into $8 \times 8$ patches. Furthermore, we random sampled 12 patches and considered these patches as input variables for each image. The remaining 52 patches are set to the baseline value.

**Implementations details.** Here, we introduce how to measure $\beta(S)$ and $\kappa(S)$ in Section 2.3. On the Tiny ImageNet dataset, we randomly sampled 100 training images. These training images were randomly sampled from different 10 classes. On the CIFAR-10 dataset, we randomly sampled 10 training images from each class. For tabular datasets, we randomly sampled 50 training samples from each class. For image datasets, we set $\tau = 2$ when the input data is normalized by its standard deviation. For tabular datasets, we set $\tau = 1$ when the input data is normalized by its standard deviation. In this way, we set the baseline $b_i = max(x_i - \tau, \mu)$, if $x_i > \mu_i$, and we set the baseline $b_i = min(x_i + \tau, \mu)$, if $x_i < \mu_i$. For Gaussian perturbation $\epsilon$, we set $\sigma = 0.02$. Besides, for each training sample, we randomly sampled five Gaussian perturbation with five different seeds, respectively.

**Adversarial attack.** Here, we introduce how to measure $A^{(s)}$ in Section 3.1. For tabular datasets, we randomly sampled 50 training samples from each class from the training set. On the Tiny-ImageNet dataset, we randomly sampled 100 training images. These training images were randomly sampled from different 10 classes. On the CIFAR-10 dataset, we randomly sampled 10 training images from each class. We used the $l_\infty$ untargeted PGD attack by following [33], in which the constraint $\epsilon = 16/255$, and the attack was conducted with 5 steps with the step size $\epsilon = 3/255$.

