# OpenReview forum: "Towards the Difficulty for a Deep Neural Network to Learn Concepts of Different Complexities"
_NeurIPS.cc/2023/Conference — NeurIPS 2023 poster_

### Official Review · Reviewer_otLW · 2023-07-05

**Soundness:** 4 excellent
**Presentation:** 4 excellent
**Contribution:** 4 excellent
**Rating:** 8
**Confidence:** 3

**Summary:**

Theoretical work using prior work defining "interactive concepts" to quantify difficulty of concepts and show why DNNs prefer learning simpler concepts by approximating the concept learning process for DNNs by linear regression.

**Strengths:**

1) Useful theoretical contribution that clearly uses interactive concept idea to formally define "easy" concepts and show why DNNs prefer to learn these shortcuts

2) Well-written and easy to follow

3) Empirical proof showing that higher order interactive concepts are more vulnerable to noise in the data

**Weaknesses:**

1) More discussion can be provided on why this particular experiment designed by the authors is a good way to verify the claims made.

2) More clear explanation of what is the contribution of prior work and that of this work.

**Questions:**

N/A

---

> ### Author Rebuttal · Authors · 2023-08-09
>
> Thank you for your great efforts. We are glad that all reviewers have given us positive comments. We will try our best to answer all your questions.
> Please let us know if you still have further concerns, or if you are not satisfied with the current responses, so that we can further update the response ASAP.
>
> ---
> Q1: More discussion can be provided on why this particular experiment designed by the authors is a good way to verify the claims made."
>
> A: Thanks. We have followed your suggestions to discuss more about the design of experiments w.r.t. the verification of our claims. Specifically, we have three main claims, *i.e.,* high-order concepts are usually less stably extracted under data noises, are learned more slowly, and are more sensitive to adversarial attacks. All these three claims have been verified in experiments, which proved the difficulty of learning high-order concepts.
>
> **For Claim 1**: *high-order interactive concepts are less stably extracted under data variations than low-order interactive concepts.* The experiment in Line 232 was designed to verify this claim in a direct manner. We directly quantified the instability metric $\kappa(S)$ of concepts of different orders. We found that high-order concepts were less stable under data variations than low-order concepts, which verified Claim 1.
>
> **For Claim 2**: *Fast learning of low-order concepts.* The experiment in Line 253 was designed to verify this claim in a direct manner. We extracted interactive concepts from the finally-learned DNN $v_{\text{final}}(x)$, and extracted interactive concepts from the DNN $v_{t}(x)$ trained after $t$ epochs. A high Jaccard similarity between $s$-order concepts extracted from $v_{t}(x)$ and $s$-order concept extracted from $v_{\text{final}}(x)$ would indicate the fast learning of $s$-order concepts, because many concepts had been already learned after $t$ epochs. Fig. 5 in the main paper shows that low-order concepts usually had higher Jaccard similarity during the learning process, which verified Claim 2.
>
> **For Claim 3:** *High-order interactive concepts are more sensitive to adversarial attacks.* The experiment in Line 285 was designed to verify this claim in a direct manner. We directly quantified the sensitivity metric $\alpha(S)$ of concepts of different orders under adversarial attacks. We found that high-order concepts were more sensitive to adversarial attacks than low-order concepts, which verified Claim 3.
>
> ---
>
> Q2: '' More clear explanation of what is the contribution of prior work and that of this work."
>
> A: We have followed your suggestions to clarify the distinctive contribution of this work over previous studies.
>
> Previous papers [2, 16, 20] mainly study the phenomenon that *DNNs easily learn simple concepts* in an experimental manner. Thus, the easy learning of simple concepts is still a common intuition without a clear theoretical formulation or analytic explanation, because how to define a concept encoded by a DNN is still an open problem.
>
> In comparison, thanks to the recent progress in [26], we follow its mathematical definition of interactive concepts. This enables us to provide an explicit theoretical connection between the complexity of concepts and the difficulty of learning concepts. Specifically, we derive an approximate instability for interactive concepts of each specific order, which reveals the high instability of high-order concepts. Thus, our research provides an approximate yet analytic explanation of the difficulty of learning high-order concepts.

---

> > ### Comment · Reviewer_otLW · 2023-08-10
> >
> > Thank you for your response, I think these explanations should be included in the revision to further strengthen the paper.
> >
> > I stand by my recommendation to accept the paper.

---

### Official Review · Reviewer_jTKY · 2023-07-06

**Soundness:** 3 good
**Presentation:** 3 good
**Contribution:** 3 good
**Rating:** 6
**Confidence:** 3

**Summary:**

This paper explores the problem of the learning difficulty of interactive concepts. The paper theoretically shows that a DNN is more likely to encode simple interactive concepts (with fewer variables in the interaction). Low-order interactive concepts are more stable to data noises and they exhibit consistent effects on the inference scores of different samples. Experiments on various DNN architectures using image and tabular datasets are done to validate the results. The paper also compares and relates their findings with existing theoretical and empirical works on trying to analyze and explain DNNs.

**Strengths:**

[S1] The paper presents a novel contribution by deriving an approximate analytical solution to the variance of interactive content's effects w.r.t data noise and showing that it increases along the order of concepts in an exponential manner.

[S2] The proof provides novel insights explaining DNNS. The findings are useful in explaining existing empirical works such as adversarial robustness results for different ordered concepts, why adversarial training is faster for certain concepts, etc

[S3] Experiments are conducted on four types of architectures and two datasets. The results are consistent with the theoretical findings.

[S4] Paper is well-written and covers various existing works.

**Weaknesses:**

[W1] Authors claim that easy samples mainly contain low-order interactive concepts. Is there existing work showing that this claim is true or can authors show it? How is "easy" defined? Is it based on how "easy" (fast?) is it for a specific network to learn or is it a more universal concept?

[W2] How was the variance value used in the experiments chosen? What is the impact if it is changed? What is the impact if a different type of noise is used?

[W3] Could authors experiment on newer architectures such as transformers? What about textual data input?

[W4] Can you provide a limitation/broader impacts section? E.g., Are there any assumptions? Does the definition of "simplicity" match real-world applicability? What can/cannot this new understanding do towards guiding the design of future networks, etc..

**Questions:**

See weaknesses.

**Limitations:**

The paper does not provide a limitations section. Perhaps authors can explain the assumptions behind the proof and its impacts on real-world applications. The paper does not provide any ethical statement in the paper nor do I think it is needed.

---

> ### Author Rebuttal · Authors · 2023-08-09
>
> Thank you for your great efforts on the review of this paper. We will try our best to answer all your questions.
>
> Please let us know if you still have further concerns, or if you are not satisfied with the current responses, so that we can further update the response ASAP.
>
> ---
> Q1:  "Authors claim that easy samples mainly contain low-order interactive concepts. Is there existing work showing that this claim is true or can authors show it? How is "easy" defined? ..."
>
> A: A good question. The claim in Line 313 that *easy samples mainly contain low-order interactive concepts* is actually supported by the heuristic findings in [4]. Cheng et al. [4] discovered that OOD samples, which were considered as difficult samples, usually contained much more high-order interactions than normal samples (simple samples). Besides, Mangalam and Prabhu [20] defined easy samples as training samples that could be correctly classified by shallow machine learning models, such as SVM. To this end, we conducted **new experiments**, in which for each $l$-th layer of an MLP, we learned a specific linear classifier $y=softmax(M * f_l)$ to use the $l$-th layer's feature $f_l$ for classification. Because high layers mainly encoded more complex features than low layers, in this experiment, we tested whether the classifier based on more complex features (in higher layers) also encoded more high-order concepts. Then, we quantified the interactions between input variables encoded by different classifiers in different layers. Fig. 1 in *the response pdf file* shows that higher layers usually encoded more complex interactions. This also partially supported the above claim.
>
> ---
> Q2:  "How was the variance value used in the experiments chosen? What is the impact if it is changed? What is the impact if a different type of noise is used?"
>
> A: Thanks. We set the variance of noise $\sigma^2=0.02^2$ in all experiments in the main paper. We have followed your suggestions to conduct **two new experiments** to answer your questions. These two experiments were conducted on the AlexNet trained on the CIFAR-10 dataset.
>
> **Experiment 1**.  We tested the instability of interactive concepts $\kappa(S)$ (defined in Line 234) by setting input noises with different variances $\sigma^2$. We added Gaussian perturbations $\epsilon \sim  \mathcal{N}(0, {\sigma}^2I)$ to each training sample. Table 1 in *the response pdf file* verifies that high-order concepts were less stable than low-order concepts. This claim was not affected by the change of $\sigma^2$.
>
> **Experiment 2**. As you requested, in this experiment, we computed the instability $\kappa(S)$ (defined in Line 234) by applying two different types of noises, *i.e.,* Gaussian perturbations $\epsilon \sim  \mathcal{N}(0, {0.02}^2I)$ and uniform perturbations $\epsilon \sim  \mathcal{U}(-0.02, +0.02)$ on each training sample. Table 2 in *the response pdf file* verifies that the type of noises did not affect the claim that high-order concepts were less stable than low-order concepts.
>
> ---
> Q3: "Could authors experiment on newer architectures such as transformers? What about textual data input?"
>
> A: Thanks. We have followed your suggestions to **conduct new experiments** on the BERT (a classical transformer-based model) on textual data. Specifically, we used the pre-trained BERT model [c1] and fine-tuned it for the sentiment classification task on the SST-2 dataset. We added Gaussian perturbations $\epsilon \sim  \mathcal{N}(0, {0.02}^2I)$ to the token embedding of each training sample. We computed the instability of the interactive concept $\kappa(S)$. Table 3 in *the response pdf file* verifies that the interaction effect of the high-order interactive concept was usually less stable than that of the low-order interactive concept on textual data.
>
> [c1] J. Devlin et al. “Bert: Pre-training of deep bidirectional transformers for language understanding,” in NAACL-HLT, 2018.
>
> ---
> Q4:  "Can you provide a limitation/broader impacts section?..."
>
> A: Thanks a lot. We will follow your suggestions to add a new section to discuss the limitation of this paper. The limitation is that there are very few ways to define and examine what is a ''concept." In this paper, we only use the following three properties to support the faithfulness of using sparse salient interactions as concepts encoded by the DNN.
> $\bullet$ [26] proved that a well-trained DNN would encode just a small number of salient interactions for inference under some common conditions. See Line 83.
> $\bullet$ [24] proved that, such a small number of salient interactions extracted from a sample $x$ could well mimic DNN’s outputs on numerous masked samples {$x_T | T\subseteq N$}. See Theorem 1.
> $\bullet$ [15] have discovered that salient interactions have considerable transferability and strong discrimination power. See Line 103.
>
> However, the above properties cannot guarantee a clear correspondence between an interactive concept and a concept in human cognition. Up to now, we cannot mathematically formulate what is a concept in cognition science. Thus, there is still a long way to unify the learning difficulty of a DNN perspective and the cognitive difficulty of human beings.

---

### Official Review · Reviewer_DenZ · 2023-07-07

**Soundness:** 4 excellent
**Presentation:** 3 good
**Contribution:** 2 fair
**Rating:** 6
**Confidence:** 2

**Summary:**

In this paper, the authors proved several theoretical results that formalized the idea that simple interactive concepts achieve a smaller variance in their interaction effects in the face of Gaussian perturbations, and are thus more stable and easier for deep neural networks (DNNs) to learn. The authors also provided experimental verifications to verify their theoretical results, and pointed out the connections between their theoretical work and previous findings on DNN learning.

**Strengths:**

- The paper provides a rigorous theoretical framework for understanding concept learning of DNNs.
- The paper formalizes the idea of conceptual complexity and establishes a formal link between conceptual complexity and learning difficulty of DNNs.
- The paper sheds light on previous heuristic findings on DNN learning, and advances our understanding of DNNs.

**Weaknesses:**

- While the theoretical derivations of the results in the paper are impressive, I am not sure how the insights from a theoretical understanding of DNN learning could help us train better DNNs. In other words, I am not certain about the practical implications and contributions of this work.

**Questions:**

How can we leverage the results and theoretical insights from this work to design better algorithms for training DNNs, or help DNNs learn complex concepts?

**Limitations:**

Since I did not find a section on the limitations of their work, the authors are encouraged to include a discussion of the limitations.

---

> ### Author Rebuttal · Authors · 2023-08-09
>
> Thank you for your great efforts on the review of this paper. We will try our best to answer all your questions. Please let us know if you still have further concerns, or if you are not satisfied with the current responses, so that we can further update the response ASAP.
>
> ---
> Q1: Ask how to use theoretical insights of this paper. "While the theoretical derivations of the results in the paper are impressive, I am not sure how the insights from a theoretical understanding of DNN learning could help us train better DNNs ... "How can we leverage the results and theoretical insights from this work to design better algorithms for training DNNs, or help DNNs learn complex concepts?""
>
> A: A good question. Analyzing the representation flaw of DNNs has become an emerging direction in recent years, *e.g.,* shortcut learning [cite1,2] and simplicity bias of DNNs [cite3,4]. For decades, researchers have realized that it is easy for an AI model to encode simple concepts, and it is more difficult to encode complex concepts. However, there are two core challenges for future development in this direction.
> (1) How to define the concepts encoded by a DNN is still an open problem. There is no formal and universally accepted definition of concepts so far.
> (2) It is still a challenge to provide analytic connections between the complexity of concepts and the difficulty of learning concepts.
> Therefore, in this paper, we theoretically explain the trend of DNNs learning simple concepts. Specifically, we prove that low-order interactive concepts in the data are much more stable than high-order interactive concepts, which makes low-order interactive concepts more likely to be encoded.
>
> The above theoretical analysis provides a specific form of the complexity of concepts that boosts the learning difficulty. Thus, in the future, we may design new neural networks, whose architectures strengthen the capacity of encoding complex (high-order) concepts and meanwhile boost the stability of these concepts (to improve the quality of concepts). However, there is still a large gap between the breakthrough in theory and achievements in practice.
>
> [cite1] Geirhos R, Jacobsen J H, Michaelis C, et al. Shortcut learning in deep neural networks[J]. Nature Machine Intelligence, 2020, 2(11): 665-673.
> [cite2] Scimeca L, Oh S J, Chun S, et al. Which shortcut cues will dnns choose? a study from the parameter-space perspective[J]. arXiv preprint arXiv:2110.03095, 2021.
> [cite3] Shah H, Tamuly K, Raghunathan A, et al. The pitfalls of simplicity bias in neural networks[J]. Advances in Neural Information Processing Systems, 2020, 33: 9573-9585.
> [cite4] Huh M, Mobahi H, Zhang R, et al. The low-rank simplicity bias in deep networks[J]. arXiv:2103.10427, 2021.
>
> ---
> Q2: "Since I did not find a section on the limitations of their work, the authors are encouraged to include a discussion of the limitations."
>
> A: Thanks a lot. We will follow your suggestions to add a new section to discuss the limitation of this paper. The limitation is that there are very few ways to define and examine what is a ''concept." In this paper, we only use the following three properties to support the faithfulness of using sparse salient interactions as concepts encoded by the DNN.
> $\bullet$ [26] proved that a well-trained DNN would encode just a small number of salient interactions for inference under some common conditions. See Line 83.
> $\bullet$ [24] proved that, such a small number of salient interactions extracted from a sample $x$ could well mimic DNN’s outputs on numerous masked samples {$x_T | T\subseteq N $}. See Theorem 1.
> $\bullet$ [15] have discovered that salient interactions have considerable transferability and strong discrimination power. See Line 103.
>
> However, the above properties cannot guarantee a clear correspondence between an interactive concept and a concept in human cognition. Up to now, we cannot mathematically formulate what is a concept in cognition science. Thus, there is still a long way to unify the learning difficulty of a DNN perspective and the cognitive difficulty of human beings.

---

> > ### Comment · Reviewer_DenZ · 2023-08-21
> > **Thank you for the response**
> >
> > After reading the response, I have revised my rating to "6: Weak Accept."

---

### Official Review · Reviewer_B4Wy · 2023-07-07

**Soundness:** 2 fair
**Presentation:** 3 good
**Contribution:** 3 good
**Rating:** 6
**Confidence:** 4

**Summary:**

The paper provides theoretical results explaining why simpler concepts are easier to learn for neural networks. The paper refers to prior work on interactive concept models, which define a concept as a subset of input features, and defines the complexity of each concept as the number of features contained within it. The main theoretical results of the paper argue that the variance of a concept and its interaction under noisy input increases exponentially with the size or complexity of the concept. Hence, more complex concepts are more likely to be influenced by small variations in the data. The paper then presents empirical evidence supporting these claims, and shows empirically that lower order concepts are more stable and learnt quicker than higher order ones.

**Strengths:**

1. The paper proposes an interesting analysis around interactive concepts.
2. The empirical evaluation backs the claims of the paper.

**Weaknesses:**

1. The theoretical analysis does not match the claims of the paper's abstract. The theory only explains higher variance for higher order concepts, but the connection between higher variance and difficulty of learning is not made explicit theoretically.
2. Several lines of work around shortcut learning and simplicity bias of networks are missing from related works.

**Questions:**

1. Can the authors make the connection between variance and learning difficulty more concrete theoretically?

**Limitations:**

See weaknesses.

---

> ### Author Rebuttal · Authors · 2023-08-09
>
> Thank you for your great efforts on the review of this paper. We will try our best to answer all your questions. Please let us know if you still have further concerns, or if you are not satisfied with the current responses, so that we can further update the response ASAP.
>
> ---
> Q1: Ask for the theoretical connection between variance and learning difficulty. "... the connection between higher variance and difficulty of learning is not made explicit theoretically."
>
> A: Thank you for your insightful comments. In fact, this is a great challenge. Up to now, there is no certificated strict correspondence between the aforementioned interactive concepts and neurons in the DNN. Therefore,  in order to derive an analytic connection between variances and learning difficulty, let us discuss our recent findings in the difficulty of learning different interactive concepts. Specifically,  we simplify the learning of a DNN into a linear regression problem. In this **very simple** setting, we assume each $i$-th concept encoded by the DNN to be a specific feature, and use $f_i$ to represent the triggering (presence) state of the $i$-th concept in a training sample. Because according to Eq. (9) in Line 200, the network output $v(\cdot)$ is the sum of a small number of salient interactive concepts, we roughly simplify and rewrite the inference output of the DNN as the following linear function
> $$y=w^\top  f=w_1f_1+\cdots +w_df_d ,$$
> where $f=[f_1,\cdots, f_d]^\top$. Then, $w_i$ can be viewed as the strength of the DNN encoding the $i$-th interactive concept. In this way, if $w_i \approx 0$, the DNN does not learn the $i$-th interactive concept.
>
> We suppose that training samples are sampled from $f \sim P(f) = N(\mu, {\Sigma}^2)$ and $y^*$ denotes the ground truth label. Specifically, $\mu=[\mu_1,\mu_2,...,\mu_d]^{\top}$ and $\Sigma={\rm diag}(\sigma_1^{2},\sigma_2^{2},...,\sigma_d^{2})$.
>
> The toy regression problem can be formulated as follows,
> $$L=E_{P(f)}[\dfrac{1}{2}(w^\top  f-y^*)^2]$$
> In this way, we can derive the optimal weights to the above regression task $|w_i| \propto |1/\sigma_i^2|$, where $w_i$ denotes the strength of the DNN encoding the $i$-th interactive concept. Please see Fig. 2 in *the response pdf file* for the proof overview (due to the limit of the page number). This is our recent finding, and we will add the full proof to the appendix if the paper is accepted. $|w_i| \propto |1/\sigma_i^2|$ indicates that the learning effect of the $i$-th interactive concept is inversely proportional to the variance of the $i$-th interactive concept. Therefore, interactive concepts with high variances (high-order interactive concepts) are more difficult to learn.
>
> In fact, it is difficult to analyze the exact dynamics of learning concepts in much more complex real-world settings. We only analyze the connection between variances and learning difficulty in the above very simplified toy setting. Nevertheless, the above analysis still provides conceptual and analytic insights into the relation between variances and difficulty of learning.
>
>
> ---
> Q2: "Several lines of work around shortcut learning and simplicity bias of networks are missing from related works."
>
> A: We will cite and discuss papers [cite1-6] for shortcut learning and simplicity bias in the related work section in the revised paper. From the perspective of the simplicity bias, our study considers the definition of interactive concepts in [26], and analyzes the bias towards learning simple concepts. More crucially, we try to induce an approximate analytic explanation from the common intuition of the bias, based on the theory of game-theoretic interactions. This work clarifies an exact form of the complexity of concepts that a DNN is difficult to learn.
>
> [cite1] Shah H, Tamuly K, Raghunathan A, et al. The pitfalls of simplicity bias in neural networks[J]. Advances in Neural Information Processing Systems, 2020, 33: 9573-9585.
> [cite2] Huh M, Mobahi H, Zhang R, et al. The low-rank simplicity bias in deep networks[J]. arXiv:2103.10427, 2021.
> [cite3] Geirhos R, Jacobsen J H, Michaelis C, et al. Shortcut learning in deep neural networks[J]. Nature Machine Intelligence, 2020, 2(11): 665-673.
> [cite4] Pezeshki M, Kaba O, Bengio Y, et al. Gradient starvation: A learning proclivity in neural networks[J]. Advances in Neural Information Processing Systems, 2021, 34: 1256-1272.
> [cite5] Scimeca L, Oh S J, Chun S, et al. Which shortcut cues will dnns choose? a study from the parameter-space perspective[J]. arXiv preprint arXiv:2110.03095, 2021.
> [cite6] Addepalli S, Nasery A, Radhakrishnan V B, et al. Feature Reconstruction From Outputs Can Mitigate Simplicity Bias in Neural Networks[C]//The Eleventh International Conference on Learning Representations. 2022.

---

> > ### Comment · Reviewer_B4Wy · 2023-08-11
> > **Response**
> >
> > I thank the authors for their response. The added proof on the toy setting as well as related works would strengthen the paper in my opinion. I am raising my rating as well.

---

### Author Rebuttal · Authors · 2023-08-09

Thanks for all reviewers' great efforts and comments. This paper has received rating of a *strong acceptance*, a *weak acceptance*, and two *borderline acceptance*. We are glad to answer all your questions and conduct **new experiments as requested.**

**Please let us know if you still have further concerns, so that we can update the response as soon as possible.**

---

### Decision · Program_Chairs · 2023-09-21

**Decision:**

Accept (poster)

**Comment:**

The reviewers were all in agreement that the paper presents interesting, novel and significant technical contributions on better understanding neural network learning and should be a good fit for NeurIPS.